# Dynamic Characteristics and Damage Detection of a Metallic Thermal Protection System Panel Using a Three-Dimensional Point Tracking Method and a Modal Assurance Criterion

**DOI:** 10.3390/s20247185

**Published:** 2020-12-15

**Authors:** Vinh Tung Le, Nam Seo Goo

**Affiliations:** 1Department of Aerospace Information Engineering, Konkuk University, Seoul 05029, Korea; vinhtung@konkuk.ac.kr; 2Artificial Muscle Research Center, Konkuk University, Seoul 05029, Korea

**Keywords:** damage detection, thermal protection system, point tracking, operational deflection shape, modal assurance criterion, dynamic characteristics

## Abstract

A thermal protection system (TPS) is designed and fabricated to protect a hypersonic vehicle from extreme conditions. Good condition of the TPS panels is necessary for the next flight mission. A loose bolted joint is a crucial defect in a metallic TPS panel. This study introduces an experimental method to investigate the dynamic characteristics and state of health of a metallic TPS panel through an operational modal analysis (OMA). Experimental investigations were implemented under free-free supports to account for a healthy state, the insulation effect, and fastener failures. The dynamic deformations resulted from an impulse force were measured using a non-contact three-dimensional point tracking (3DPT) method. Using changes in natural frequencies, the damping ratio, and operational deflection shapes (ODSs) due to the TPS failure, we were able to detect loose bolted joints. Moreover, we also developed an in-house program based on a modal assurance criterion (MAC) to detect the state of damage of test structures. In a damage state, such as a loose bolted joint, the stiffness of the TPS panel was reduced, which resulted in changes in the natural frequency and the damping ratio. The calculated MAC values were less than one, which pointed out possible damage in the test TPS panels. Our results also demonstrated that a combination of the 3DPT-based OMA method and the MAC achieved good robustness and sufficient accuracy in damage identification for complex aerospace structures such as TPS structures.

## 1. Introduction

A thermal protection system (TPS) is necessary for hypersonic aircraft and spacecraft to withstand aerodynamic heating and acoustic loads during hypersonic flight. The TPS panel serves as a shielding cover for the fuselage of the vehicle and endures possible impact damage from debris [1,2]. Several types of TPS, such as metallic TPS [3], multilayer TPS [4], integrated TPS [5], and bioinspired TPS [6], have been proposed and investigated through the thermal [7,8,9] and thermo-mechanical performance [10] tests.

A vehicle’s fuselage can be exposed to more extreme thermal, pressure, and impact loads if a TPS panel becomes damaged [11,12,13,14]. Thus, the TPS panel must be in good condition before launch because of its critical role in protecting the vehicle’s structures, subsystems, and even humans. Understanding the structural response and damage state of the TPS panel is important in design and maintenance processes. Chen et al. [15] presented an analysis method to study the panel flutter of a metallic TPS. They calculated natural frequencies and predicted critical dynamic flutter. They concluded that the outer sandwich of the metallic TPS panel was not susceptible to panel flutter but the panel-to-panel seals of the metallic TPS panel were susceptible to panel flutter. Tobe et al. [16] developed a method for localizing fastener failure and material optimization in TPS panels. They employed accelerometers to measure dynamic response of TPS panels under an impulse force. However, due to restrictions of the experimental apparatus, the dynamic characteristics of the TPS panel could not be sufficiently considered and their damage investigation of the TPS panel was limited. Since the inspection of the TPS panel by traditional methods is laborious and expensive [17], what remains to be developed is a new and simple inspection method that has high accuracy and less inspection time. The purpose of this paper is to present a structural health monitoring (SHM) method which improves the performance of conventional methods for investigating the damage state of the TPS panel in the laboratory.

We designed metallic TPS panels to be connected to the vehicle’s fuselage via mechanically bolted joints that provide functions of structural connections, energy dissipation, and vibration damping in a metallic TPS panel. Figure 1a,b shows an example of TPS panels with fasteners as structural connections at four corners of the panel. Damage modes of bolted joints such as self-loosening, fatigue, and separation can lead to the penetration of hot air into the interior of the vehicle, which might result in the catastrophic loss of the vehicle. In our previous study [10], a metallic TPS panel was investigated in a thermomechanical experiment and simulation. Figure 1c shows a picture of the current metallic TPS panel. We found that after several simulated missions, there was a permanent deformation of the metallic TPS panel that was due to the plastic deformation of the bolted joint structures, which resulted in the reduction of the clamping forces of the fasteners. Consequently, the TPS panel would be vulnerable to deflection and damage under vibration loads in the next mission, and thus the goal of TPS reusability would not be achieved. Therefore, the damage due to bolted joint failure is the main concern in our current research.

One of the traditional methods to detect damage employs visual inspection. This method is adequate if the damage is visible on the surface of the TPS panel, such as cracks, holes, and fractures. However, if the features are not visible on the surface, such as damage to a bolted joint or damage in interior components, then visual inspection is not sufficient. The primary methods used to detect bolted joint damage or invisible damage usually utilize the dynamic characteristics of the structure. Damage to structures causes changes to the modal parameters of the structure, such as the natural frequency, damping ratio, and mode shape, compared with the initial or fully healthy state. Therefore, bolted joint damage in structures also causes changes in the dynamic characteristics [16]. Conversely, we can detect the damage or failure by identifying the changes in the modal parameters, such as the natural frequencies [20] and the damping ratios [21].

Along with changes in natural frequencies and damping ratios, a change in mode shapes is also considered for damage detections. The mode shapes are less sensitive to environmental effects, such as temperature, than natural frequencies [22]. The modal assurance criterion (MAC), which indicates changes in mode shapes, has commonly been used for model validation [23]. Orlowitz et al. [24] used accelerometers mounted on a square-plate structure to measure the structural dynamics. A large number of predefined points on the structure were used to obtain mode shapes, and MAC values were compared to each other to validate the proposed experimental methods. Nguyen et al. [25] used a laser vibrometer to obtain mode shapes of a composite disc from a large number of predefined measurement points and calculated MACs for damage detection. The use of the MAC in damage detection requires detailed mode shapes (many points) and higher frequency modes to obtain better results [26]. Other considerations of damage detections are the Modal Curvature method (MCM) [27] and Modal Flexibility method (MFM) [28]. Pandey et al. [29] presented a damage detection method based on changes in mode shape curvature which was known as the MCM. The curvature values were computed from the displacement mode shape using the central difference operator. It assumed that damage-associated stiffness reduction increased the curvature. The damage localization can be determined by evaluating the largest computed MCM value. This methodology demonstrated a high level of damage sensitivity. However, the MCM also presented some drawbacks; one of them is errors due to the application of the central difference approximation method to displacement mode shapes [30]. It might result in a false damage localization [31]. The MCM alone is not recommended for damage identification, it may be used in conjunction with other sub-optimal modal parameters [32]. Later on, Pandey and Biswas [28] proposed an approach for damage detection based on flexibility change of the structure. The MFM defines the flexibility matrix as the inverse of the stiffness matrix. The flexibility matrix could be determined with fewer modes than was required for the stiffness matrix. Estimation of damage locations based on the modal flexibility method depends on the number of sensors used and distance of the sensors to the damaged location.

Several experimental methods have been used for damage detection of structures: accelerometers [16,33], laser displacement measurement [34,35], photogrammetry [36,37,38,39], infrared thermography [40], and ultrasound [41]. Among them, the method that uses accelerometers has the effect of mass loading for lightweight structures and a labor-intensive and time-consuming process for large structures. In addition, the contact measurement is pointwise and limited to only a few locations. These limitations have been solved by recent progress in non-contact measurement methods, which are able to measure displacements or temperature without invading the structures [34,40,42]. In the non-contact method using infrared thermography, the structure being heated detects cracks or damage, while the non-contact method using ultrasound is also suitable for cracks or damage in small-scale structures. Moreover, the non-contact method using a laser has difficulty with large displacement measurement or the presence of large rigid body motions and needs much acquisition time for a large number of measurement points. In addition, the full-field mode shape measurement requires much inspection time in these existing contact and non-contact methods.

The non-contact method using photogrammetry along with vibration tests is preferred for damage detection based on the changes in the dynamic characteristics of structures [43]. By tracking movements of exterior features of a structure that appear on digital images captured by cameras, the displacement can be calculated by photogrammetry theory and its modal parameters can be estimated for damage detection. The digital image correlation (DIC) method is a kind of photogrammetry method that measures the full-field deformation of structure in various fields of research, such as for vibration [44], crack monitoring [45,46,47], high-temperature structures [48,49,50], electronic packaging [51,52,53], and small structures [54,55]. The identification of a whole area displacement field or operating deflection shape (ODS) is one of the most interesting features using the DIC method [56]. Thus, the use of the DIC method in vibration and modal analysis has become common [57]. Wu et al. [58] used the DIC method to measure displacements and strains in rotating wind turbine blades, and they used the blade displacement in the frequency domain to identify faults. Helfrick et al. [59] used the DIC to detect damage based on changes in the curvature of the structure’s displaced shape. The damages were created as cracks presenting in a beam structure.

Moreover, the three-dimensional point tracking (3DPT) method is also a kind of photogrammetry method that uses a pair of digital cameras to measure 3D displacements of discrete points attached to a structure [60]. The 3DPT uses removable optical targets attached to the surface of a structure. The removable optical targets make the 3DPT method preferable if the structures in service are inspected and require decontamination to the test structure. A TPS panel assembled on a hypersonic vehicle is an example. The use of the 3DPT method in vibration analysis has become very popular, and both laboratory vibration measurements and large-scale outdoor measurements have been reported [60,61,62]. Poozesh et al. [60] successfully tested and validated the 3DPT method in a large-area measurement of wind turbine blades. Warren et al. [61] compared the mode shapes of a base-upright structure obtained from accelerometers, 3D laser vibrometers, full-field DIC, and 3DPT methods. MAC values were determined to validate the measurement methods. They reviewed similar results from each method and showed that the 3DPT method was the most suitable method for measuring the dynamic characteristics of a structure. In addition, for a typical application that requires a small amount of data storage and short computation time, the 3DPT is preferable over the DIC method because the measurement of the discrete points has a smaller amount of data storage [61]. By attaching multiple points on a testing structure, the 3DPT becomes a special laser vibrometer measurement method that can simultaneously measure responses of predefined points. Note that any signs of damage that cause changes in modal parameters are important in the TPS panel. When a sign of the damage was detected, the TPS panel would be inspected immediately by removing the TPS panel from the vehicle’s structure. Therefore, we only focus on the damage existence in the TPS panel rather than localizing exactly the position.

From the above-mentioned methods with 3DPT and MAC, we find that (1) using 3DPT for detecting damage in a structure based on dynamic measurement, we can obtain more data compared with the traditional pointwise measurement method, and data storage and computation time are reduced compared with the full-field DIC method but precise mode shapes of the testing structure are still guaranteed. (2) Using MAC for damage detection, we can detect changes in global mode shapes of a testing structure. MAC is well suited for modal analysis of large-scale structures using a simultaneous measuring response method. Therefore, we employed the 3DPT method in conjunction with the MAC, which is a non-contact image-based method, to investigate dynamic characteristics, to obtain detailed mode shapes, and to identify damage states in the metallic TPS panel that are associated with the bolted joint loosening.

In this paper, we present a highly efficient 3DPT method in a conjunction with the MAC to investigate the dynamic characteristics of a complex structure; that is, a metallic TPS panel that consists of bolted joints, insulation layers, a load-carrying plate, and washers. We focus on detecting damage existence in the TPS panel rather than localized damage. Three damage levels represented three cases of the number of damaged corners. For the experiment, we provided a discussion of good selections of the aperture and suspension system by quantifying rigid body motions and rigid body natural frequencies. Then, an operational modal analysis (OMA), or ODS measurement during modal tests, was performed. The modal parameters, namely, the natural frequency, mode shape, and damping ratio, were identified with the aid of the OMA. The results of the 3DPT method have also been validated using the benchmark modal data from the accelerometers. The damage states were detected by comparing the natural frequencies and damping ratios with those of a healthy TPS panel. By taking advantage of the simultaneous multi-point measurement data, the ODSs were constructed and the modal matrix of the TPS panel was obtained from each measurement. The modal matrices of the first four modes were then processed to obtain MAC values which were used to identify the damage states. The signal-to-noise ratio of the captured images was too small in a certain area because of the limited resolution of cameras, intrinsic noise, low impact energy due to localized excitation, and infinitesimal deformation at the high frequency of the TPS panel. To remedy this drawback, a phase-based motion magnification was used. The results obtained from the 3DPT method demonstrated the advantage of using a non-wired setup and simultaneous measuring responses over the use of a single measuring response by an accelerometer. By using simultaneous multi-point measurement data for the MAC calculation, the 3DPT method reduced processing times significantly in the damage detection of a complex structure.

## 2. Theoretical Background

This section presents the theoretical background of our study. Equation (1) is the governing differential equation of a multi degree-of-freedom system (multi-DOF) for the forced vibration of a structure in a matrix form.
(1)$$\left[M\right]\left\{\ddot{x}(t)\right\}+\left[C\right]\left\{\dot{x}(t)\right\}+\left[K\right]\left\{x(t)\right\}=\left\{f(t)\right\},$$
where *M*, *C*, and *K* are the mass, damping, and stiffness matrices, respectively; *x* and *f* are vectors of the displacement and load, respectively.

The frequency response function (FRF) at a specific frequency (*jω*), *H(jω)*, is calculated by dividing the spectrum of output to input signals. For a certain degree of freedom *p* and an excitation force at the *q*-th degree of freedom, the FRF is defined as *H_pq_* = *X_p_*/*F_q_*.

The frequency-domain ODS is simply defined as the forced response at a specific frequency (*jω_0_*), as shown in Equation (2). We assumed the linearity of the structure to obtain the ODS. Equation (2) shows that the component of an ODS vector is defined as the multiplication of the row of FRFs corresponding to the excitation DOF and the Fourier transform of an excitation force.
(2)$$\left\{ODS(j\omega_{0})\right\}=\left[H(j\omega_{0})\right]\left\{F(j\omega_{0})\right\}$$
where *F*(*jω_0_*) is the force in the frequency domain at a specific frequency (*jω_0_*).

It is clear that the ODS is dependent on the applied excitation force. However, in the field of OMA, the frequency response function cannot be obtained since the applied excitation is not available. Therefore, the ODS FRF or transmissibility function can be used to obtain modal parameters in the OMA [63,64]. The ODS FRF under an excitation condition between the output and the reference output (*X_ref_*) was defined as the ratio between the two responses *X*(*jω*) and *X_ref_*(*jω*).

The ODS can be obtained directly by extracting the relative magnitude and phase from a set of ODS FRF measurements at a frequency of interest. Every row or column of the FRF matrix contains the same mode shape and its components. Thus, we only need one row or column of the FRF matrix to measure the modal parameters of the structure. In this study, we used two experimental methods. The first one was the roving excitation force and fixing response (single accelerometer) and the second one was a simultaneous multi-response measurement and the fixing excitation force. The experimental methods demonstrated the advantage of using a simultaneous measuring response (3DPT) over the use of a single measuring response (accelerometer).

Estimates of the modal parameters from the measured ODS FRFs were obtained by curve fitting ODS FRFs data [65,66,67] using the ME’scope^TM^ software (Vibrant technology, Inc.) for the roving excitation and using in-house Python code in 3DPT-based ARAMIS^®^ software [68] for the simultaneous measuring response. Under a forced vibration, the structure deforms as a combination of multiple modes. For structures that have low values of damping, the modal frequencies are far apart from each other and the frequency at the resonance peak in the ODS FRF is taken as the modal frequency. However, for structures that have high values of damping, the ODSs contain many closely spaced modes or much participation from neighboring modes [69,70]. In this study, we considered the modal frequencies only to be obtained from curve fitting FRFs of a single predefined DOF under a particular impact point. Thus, the closely spaced modes may make one averaged mode in the band of curve fitting.

## 3. Material and Methods

### 3.1. Material Preparation

The metallic TPS panel used in the experiments consisted of an Inconel 625 for the front plate, a titanium Ti-6Al-4V alloy for the back structure, bolted joints made of stainless steel S304, spacers and washers, and thermal fibrous insulation material. The nominal dimensions of the TPS panel are 170 mm × 170 mm × 29.54 mm. The front plate with an overhang at four edges is called a front load-carrying plate. The size of this front load-carrying plate was 240 mm × 240 mm × 2.54 mm. The overhang made an overlapping area between adjacent TPS panels, which helped to prevent an inflow of hot air. Figure 2 shows the metallic TPS panel with components and detailed dimensions. Table 1 shows the size and weight of the components in the TPS panel.

In this study, we collected three cases of experimental data which included the tests of a single load-carrying plate (Inconel plate), a metallic TPS panel without thermal fibrous insulation material, and a fully assembled metallic TPS panel. First, the single load-carrying plate was tested because this plate was one of the most important components of the TPS panel which was dominant in the weight of the TPS panel. We investigated the dynamic characteristics of this plate and the relationship of its dynamic characteristics to those of the TPS panel. The second case was for the assembled TPS panel without fibrous insulation material, called the skeleton of the TPS panel. This was to show how much damping was involved without the addition of the fibrous insulation material (nonstructural mass). The third case was for the fully assembled TPS panel which was divided into four test sub-cases to study dynamic characteristics of the TPS panel under healthy conditions and damage conditions at specific bolted joints.

A photo of the test TPS panel is shown in Figure 3a. To make the healthy condition, a torque wrench with a torque value of 4 N·m for the bolted joints was used, as shown in Figure 3b. Note that the bolt type used in the TPS panel was M5 with a grade of 6.8 which was recommended to tighten at a torque value of less than 4.56 N·m [71]. The damage state was made by loosening a bolted joint, as shown in Figure 3c. The three damage states of the bolted joint were defined as a bolt loosened at corner 1, bolts loosened at both corners 1 and 3, and bolts loosened at both corners 1 and 4. The details of experimental cases are listed in Table 2.

### 3.2. Experimental Setup

Figure 4 shows the experimental setup for dynamic tests using a high-speed 3DPT method and a roving method with an accelerometer. Two high-speed cameras (Photron FASTCAM APX RS) with a resolution of 1024 × 1024 pixels and 50 mm focal lengths (Nikon AF NIKKOR) were used to capture the vibration of the TPS panel. The rigid body motion generated by an impulse force (impact hammer) under the free-free boundary condition was designed to be no larger than 100 mm so that the motion was within the field of view in the measurement of out-of-plane deformations. In this measurement frame rate was selected as 3000 frames per second, the shutter speed was set at 1/6000 of a second, and an aperture size of f/11 was selected. Optical targets were attached to the front surface of the test structure in such a way that the overall shape formed by the attached targets could represent the shape of the test structure. Two high-speed cameras were set on a tripod stand with a built-in spirit level to ensure a horizontal level. Before starting the measurement with the 3DPT method, the camera system was calibrated to identify the distance and the angle between the two cameras.

According to the practices guide [72], for the DIC provided by the international digital image correlation society, the stereo angle of the two cameras with the current focal lengths was set at 15 degrees to reduce out-of-plane uncertainty. In the calibration result, the working distance was 950 mm. The calibration deviation was determined as 0.015 pixels, which corresponded to 0.004 mm for the measurement volume of 270 mm × 270 mm × 270 mm. The depth-of-field of the calibrated system (270 mm) guaranteed a good focus for the rigid body motion limit of 100 mm. The spatial resolution was calculated as 0.26 mm/pixel. The sensitivity of the out-of-plane measurement was approximately 1/30 pixels, which corresponded to roughly 10 µm for this test [73]. The noise floor was estimated by taking a series of 100 images at rest. The average filter was applied to reduce the noise in the measurement system, and Figure 5a shows that the noise floor was mostly below 35 µm. Maximum and minimum displacements in a measurement, including self-deformations and rigid body motions, are shown in Figure 5b. The rigid body motions, which ranged from −30 to 90 mm, were well within the depth-of-field of the calibrated cameras. Note that the selected aperture f/11 was appropriate to prevent motion blur. Figure 5c shows the impulse force history and its autospectrum. After several trials, we found that the impulse force should not exceed a certain value (less than 200 N) to guarantee the rigid body motion limit of 100 mm.

The camera system was placed to face the front surface of the test structure on which the optical targets were attached. Two fine-nylon bands were used to suspend the test structure vertically to make a free-free boundary condition. Three rigid body natural frequencies of the system, 4 Hz, 11 Hz, and 17 Hz, were obtained from the experiment. Note that these three frequencies satisfied the requirement of free-free support conditions, which was no more than one-tenth of the frequency of the lowest elastic mode (192 Hz as presented in Section 4.1) [74,75,76]. Therefore, the use of two fine-nylon bands for the suspension system in this study was sufficiently soft for reducing the errors of stiff supports in the measured modal frequencies. A photoelectric sensor was placed right behind the test structure so that an electrical signal was sent out to trigger the cameras when the impact hammer passed the light transmitter of the photoelectric sensor, as shown in Figure 4. Note that the impact point was chosen at the overhang area of the front load-carrying plate, as shown in Figure 4. If the impact point was on the back of the structure (titanium plate), the fibrous insulation material of the TPS structure would absorb the impact energy and the test structure would not be excited well. The cameras captured the images of the test structure, which included several images prior to the impact and numerous images during impact. The captured images were then processed with the 3DPT-based ARAMIS^®^ software to obtain the out-of-plane deformation. The measured deformation from the 3DPT method was then transferred from the time domain to the frequency domain to obtain the ODS FRF functions of the test structure. The spectrum averaging technique and Savitzky–Golay smoothing method were used to reduce the noises of signals.

At the same time, we also used an accelerometer to measure the response of the test structure. The roving impact hammer method was used to obtain the FRFs at predefined points on the test structure [63,69,77,78]. An accelerometer was attached at a fixed point near the corner, and the impact hammer was used to excite the TPS panel at marked points to define its mode shapes. We used 36 marked points and 25 marked points in the load-carrying plate and the back structure (titanium), respectively. Figure 6 shows the schematic and the numbering system of the marked points in the test structure. The signals from both the accelerometer and the excitation force of the impact hammer were then processed in both B&K Pulse^TM^ and ME’scope software. One FRF between each impact point and the fixed response point was computed with five spectrum averages and an exponential window for a measurement frequency span of 800 Hz and 800 frequency lines. The 61-time impact made a full FRF matrix.

In the theory of experimental modal analysis (EMA), the modal parameters, the natural frequency, damping ratio, and mode shape of the test structure, were obtained from the FRF, which is defined as the complex ratio of the output to the input. Because of the limitation of the current hardware, the input signal from the impact hammer and the output response from the high-speed cameras were not sampled simultaneously. Therefore, the FRF could not be obtained and the OMA was used instead. This method used the output response to characterize the modal parameters of the test structure. The impact force was assumed to be the unknown excitation force and the measured response of the test structure was used to compute the ODS FRF and to construct the ODS [79]. This new type of measurement is called ODS FRF measurement [33,69,78] or transmissibility measurement [77,80]. If the ODS at or near a resonant frequency would be dominated by a single corresponding mode shape at that resonant frequency, then the ODS could be approximate to the mode shape [63,66,78]. Therefore, in this study, we used the ODSs from the ODS FRF measurement to compare them with the mode shapes from the roving hammer method and the mode shapes from the numerical simulation method in the validation.

A drawback of the displacement-based measurement method, such as the 3DPT method, is that when a structure is excited with only a small amount of energy, the structure may experience such a small deformation that it is not perceptible by the raw data of the 3DPT method. This can be explained by a small magnitude of displacement at high frequencies, and hence it misses a natural frequency corresponding to a particular mode. After many impact trials in this study, we determined the optimal impact point to be a point in the overhang area on the back surface of the front load-carrying plate (Inconel), as shown in Figure 4. Since the amount of excitation energy was not enough to drive the whole structure, a certain mode was not excited much and was not seen clearly in the original ODS FRF curve. The problem of undiscovered modes was also mentioned in Reference [24]. For a complex geometry such as the proposed metallic TPS panel in this measurement, it was difficult to extract all the modes from the OMA test because of a localized excitation point in the OMA laboratory test [81]. A method to find the undiscovered modes is necessary.

The study on the vibration measurement using the 3DPT and phase-based motion magnification method has been widely adopted to identify the modal parameters and structural dynamics of structures. Wadhwa et al. [82,83] developed a process of phase-based video motion and published a MATLAB-based program [82]. Chen et al. [84] first used the phase-based motion magnification to identify the modal analysis of simple structures. They demonstrated the algorithm’s capability of identifying the ODSs of a cantilever beam from a video motion and compared the results with those measured by a laser vibrometer. With that they confirmed the accuracy of the method and concluded that the phase-based motion magnification would be readily adaptable to the SHM of structures in real applications. Poozesh et al. [85] and Sarrafi et al. [86] successfully used the phase-based motion magnification technique with small amplitude tests, including the use of small exciters or high frequency, to identify the ODSs of a vibrating structure that could not be determined by a traditional photogrammetry method. Civera et al. [87] used the phase-based motion magnification to study damage detection and localization in ODSs. The results showed that model-based SHM could be performed on modal data with phase-based motion magnification technique.

In this study, the signal-to-noise ratio in the series of the captured images was improved using the phase-based motion magnification, in order to use the 3DPT method to identify dynamic characteristics of the metallic TPS panel in the above-mentioned conditions. The motion of the captured images that had unclear peaks in the ODS FRF curve was magnified by a certain magnification factor.

We used a phase-based motion magnification technique for a case study of a healthy fully-assembled TPS panel. A magnification factor was applied to a specific bandwidth of the frequency domain. The algorithm decomposed the input signals into the local spatial amplitude and phase signals using complex steerable pyramid filters. After the reconstruction of images, a higher amplitude at the magnified frequency band was achieved. The requirements of the magnification parameters are as follows: frame rate, magnification bandwidth, and magnification factor. Figure 7 shows an overview of the combination of motion magnification and the 3DPT method to obtain magnified displacements of the TPS panel.

### 3.3. Validation

We needed to validate the proposed measurement method for the test structure using the OMA-based 3DPT method with the dynamic characteristics of the load-carrying plate, and we also performed a finite element analysis of the load-carrying plate. Additionally, we included the results of the dynamic characteristics from the accelerometer measurement. The comparison in the natural frequencies and mode shapes between the measurements (3DPT and roving hammer) and the finite element analysis are shown in Figure 8 and Figure 9.

Figure 8a shows the deformation responses of the load-carrying plate in the time domain measured by the 3DPT method. A fast Fourier transform (FFT) was performed on the measured responses of the load-carrying plate. Figure 8b shows the ODS FRF results of a single FFT, the averaged results of 20 ODS FRFs, and the smoothing curve of the averaged result (Savitzky–Golay smoothing method with an order of three and a window width of five measured by the 3DPT. There are four peaks on the ODS FRFs of the load-carrying plate using the 3DPT method. Figure 8c also shows similar values of peaked frequencies that exist on the FRF curves of the load-carrying plate measured by the roving hammer method. There is good agreement of measured frequencies between the 3DPT method and the roving hammer method.

An eigenvalue frequency analysis of the front carrying-load plate was carried out in ABAQUS finite element analysis (FEA) software. C3D8R elements with eight-node linear brick and reduced integration were used. There were 78,148 elements in total. The measured response of optical targets was then used to obtain the ODSs of the load-carrying plate by an in-house Python code. Because the ODSs were not scaled by the impact force, they were also called unscaled mode shapes or operational shapes, as shown in Figure 9a. Note that the contour color defined in the ARAMIS^®^ software follows the scale from −1 to 1 while it is an absolute scale from 0 to 1 in the ME’scope^TM^ software (Figure 9b) and FEA (Figure 9c). In the overall results, the natural frequencies and mode shapes from the 3DPT method were similar to those from the roving hammer method and the finite element analysis. Hence, the current 3DPT method achieves acceptable accuracy and is appropriate for investigating the characteristics of the TPS panel throughout this study.

### 3.4. Damage Detection Method

We used two methods to evaluate the damage states of the TPS panel. The first method was based on the change in natural frequencies and damping ratios. The other method evaluated the mode shapes based on the MAC. In this study, among many damage states of the TPS panel, bolted joint loosening was critical for the safety of a vehicle. Loose bolted joints also made significant changes in frequencies, damping ratios, and mode shapes of the metallic TPS panel in most test cases. We focused on this damaging problem to investigate the applicability of our damage detection method.

It is a very important but difficult to define the threshold of a damage state because there is a measurement uncertainty due to noises of acquisition devices and environmental conditions. This measurement uncertainty causes a small change in frequencies, damping ratios, and mode shapes. Therefore, an appropriate experimental setup is needed to minimize noise. We assumed that the camera system was well calibrated and the optical targets had good contrast. Three factors, namely, lighting, camera positioning, and optical targets, were regulated until most of the noise floor was within the desired accuracy. The temperature of the environmental ambient was also maintained at room temperature. The maximum noise floor was 35 µm while the maximum deformation response of the TPS panel was approximately 300 µm. This represented a maximum measurement uncertainty of 11.6%. For a single TPS panel, we used this threshold for the damage detection.

The MAC provides a measure of consistency between estimates of a mode shape. This method compares the mode shape vectors between healthy and damaged cases and gives a MAC value for each mode. If the two-mode shapes are the same then the MAC value will be one; otherwise, it will be zero if they are completely different. The mathematical representation of the MAC is expressed in Equation (3).
(3)$$MAC{\left(u,d\right)}_{r}=\frac{{\left|{\left\{\psi_{ur}\right\}}^{T}\left\{\psi_{dr}\right\}\right|}^{2}}{{\left\{\psi_{ur}\right\}}^{T}\left\{\psi_{ur}\right\}{\left\{\psi_{dr}\right\}}^{T}\left\{\psi_{dr}\right\}},$$
where *ψ_ur_* is an undamaged mode shape, *ψ_dr_* is a damaged mode shape, the subscript *r* represents the *r*-th mode, and the superscript *T* indicates the transpose of a vector.

The flowchart of the proposed damage detection method using a combination of the 3DPT and MAC methods is shown in Figure 10. The deformation response of predefined DOFs (optical targets) in a case study was exported from the 3DPT-based ARAMIS^®^ software. The spectrum data for the deformation response was calculated using the FFT algorithm in MATLAB program. The natural frequencies were at peaks of the ODS FRF curves and the damping ratios were calculated by the half-power method. Modal vectors were calculated by examining the imaginary parts of the ODS FRF. The modal vector of the *r*-th mode was scaled with the maximum value among predefined DOFs of the *r*-th mode. The MAC matrix was then obtained from Equation (3) to evaluate the consistency of two case studies. The damage detection was then identified based on changes in natural frequencies, damping ratios, and mode shapes.

## 4. Results and Discussion

### 4.1. Dynamic Characteristics of a Healthy Fully-Assembled TPS Panel

This section presents the dynamic characteristics of a healthy fully-assembled TPS panel measured by the roving hammer method. Figure 11 shows the FRF curve of the measurement response at point 16 on the front surface of the healthy fully-assembled TPS panel. The peaks in the FRF curve are clearly identified. The first four natural frequencies were 192 Hz, 330 Hz, 361 Hz, and 464 Hz

We used the FRFs data of 61 marked points on both the load-carrying plate and the back structure (titanium) to construct the first four mode shapes of the healthy fully-assembled TPS panel. ME’s Scope software provides the mode shapes corresponding to the natural frequencies, as shown in Figure 12. The mode shapes obtained from the proposed metallic TPS panel under the free-free condition agreed well with those from the free-free vibration of a sandwich panel in Reference [88]. The mode shapes in Figure 12 confirmed a similarity to those of the load-carrying plate (Figure 9) because of the dominant weight of the load-carrying plate in the TPS panel.

The TPS panel twisted and formed the torsional mode for the first mode and showed the bending modes for the second and third modes. The fourth mode was a kind of coupled bending-torsion mode which contained both a bending component and a twisting component.

Next, we measured the dynamic characteristics of the healthy fully-assembled TPS panel by the 3DPT in the simultaneous measuring response (optical targets). Figure 13 shows the response of the points of interest on the front surface of the TPS panel. As discussed earlier, for the impact point on the TPS panel, the impact hammer drove an amount of energy that could not generate significant deformation at the central area of the front plate of the TPS panel. Therefore, there was only a small deformation response at the center area (point 2) while there was a larger deformation response at the edge area (point 1).

The spectrum of the deformation response (point 1) was obtained by an FFT with a uniform window function. Note that the use of an exponential window function in the FFTs might add artificial damping to the measurements [89]; therefore, we decided to use the uniform window function in the FFTs because the most important information on deformation responses was in the initial part of the time record (within 0.3 s). Figure 14 shows the single spectrum of the response and the averaged spectrum of 20 ODS FRFs. There are three peaks, 192 Hz, 332 Hz, and 467 Hz, in the ODS FRF curve. A small unclear peak at 362 Hz showed that there might be a natural mode at that frequency but it could not be identified clearly. A possible reason is that the deformation obtained by the 3DPT method at the central area of the TPS panel was very small (signal-to-noise was about 1), so the peak was not recognized clearly at 362 Hz in the 3DPT measurement method. Note that there was a bending mode at 361 Hz measured by an accelerometer, where the most deformation was exhibited at the center area of the TPS panel. Generally, an accelerometer is more sensitive than any deformation measurement sensor since the acceleration is a frequency square order bigger than the deformation. This is a drawback of the displacement-based measurement method in measuring dynamic characteristics of complex structures if the deformation is small due to localized excitation.

The series of the captured images of the vibration test was processed to magnify the motion using a phase-based magnification method, as discussed earlier. The frame rate was 3000 frames per second, the magnification factor was three times, and the frequency band was from 355 Hz to 365 Hz. Then the magnified images were processed with the 3DPT algorithm to obtain the magnified displacements. The result of the new spectral data after the motion magnification is shown in Figure 15. The signal-to-noise ratio increased significantly. The peak at 362 Hz was shown clearly in the new spectral data. The ODSs of the healthy fully-assembled TPS panel are shown in Figure 16. The natural frequencies and corresponding ODSs obtained from the 3DPT method and the motion magnification agreed well with those from the accelerometer measurement method.

The fibrous insulation material was removed from the healthy fully-assembled TPS panel to study the effect of the fibrous insulation material on the structural dynamics of the TPS panel. A torque value of 4 Nm was maintained. The results of the dynamic test of the skeleton TPS panel are shown in Figure 17. The deformation of a point of interest on the front plate of the TPS panel is shown in Figure 17a. The spectrum data of the deformation response are shown in Figure 17b. Obviously, the deformation history of the skeleton TPS panel without the fibrous insulation material lasted longer than that from the healthy fully-assembled TPS panel with the fibrous insulation material (Figure 13c) because of the damping effects of the fibrous insulation material. This also resulted in a clear peak at the third natural mode (363 Hz), as shown in Figure 17b. The natural frequencies of the skeleton TPS panel increased correspondingly from the TPS panel with the fibrous insulation material. Because the fibrous insulation material was considered to be a non-structural mass, the increase in the natural frequencies was due to the decrease in the mass of the skeleton TPS panel without the fibrous insulation material.

### 4.2. Damage Detection

Figure 18 shows the ODS FRF curves of the TPS panel with damage at specific corners. In general, the natural frequencies of the damaged TPS panels changed from the healthy fully-assembled TPS panel. Specifically, the natural frequencies decreased due to the degradation of the rigidity of the TPS panels as they were damaged. The degradations increased with an increase in the number of damaged corners. Figure 18a,c,e shows the deformation responses of the TPS panel with damage at corner #1, at both corners #1 and #3, and at both corners #1 and #4, respectively. Because of a single damaged corner, the TPS panel with damage at corner #1 had higher rigidity than the other two cases (damage at both corners). This resulted in higher natural frequencies of the TPS panel with damage at a single corner, as shown in Figure 18b,d,f. Moreover, we can see that the damage at both corners #1 and #3 (two bolted joints in the diagonal line) resulted in a high rigidity of the TPS panel compared to the damage at both corners 1 and 4 (two bolted joints in the vertical line). A summary of the changes in natural frequencies of the test structures and the percent differences from the healthy condition are listed in Table 3. The frequency changes were significant. Therefore, changes in the natural frequencies can be used to detect damage to the TPS panel.

The calculated spectrum curves of the test structures were fitted with the smoothing technique. The smoothing curves were plotted along with the calculated spectrum curves. Modal damping was also estimated from these smoothing spectrum curves. The half-power bandwidth method was used to calculate the damping ratio. The effect of the fibrous insulation material was described by considering the damping ratio of each test case. The bandwidth was calculated using a 3 dB cut-off frequency, and the damping ratios of the first natural mode of each test case are listed in Table 4. As we can see from the damping ratios, the single load-carrying plate had the lowest damping ratio. The TPS panel with the fibrous insulation material had a higher damping ratio than the TPS without the fibrous insulation material because of the damping ability of the fibrous insulation material.

The damping ratios of the damaged cases were higher than those of the healthy fully-assembled TPS panel because the thickness of the insulation layer increased (increase of porosity) as the damage occurred (loosening of the bolted joints), which resulted in the high damping capacity of the fibrous insulation material [90]. The damping ratio of the damaged TPS panels at a single corner was lower than that of the TPS panels at two damaged corners. The damping ratio of the TPS panel with damage along the diagonal line (corners #1 and #3) was lower than that of the TPS panel with damage along the vertical line (corners #1 and #4) because the damage along the diagonal line still guaranteed the compaction of the fibrous insulation material while the greater expansion of the insulation layer occurred in the damage along the vertical line. These results indicated that changes in damping ratios can also be used to detect the damage states of the TPS panel.

The ODSs of the test cases are summarized in Figure 19. The ODSs of the test cases were clearly related to the ODSs of the load-carrying plate. The ODSs of the healthy fully-assembled TPS panel with or without the fibrous insulation material were almost the same as the ODSs of the load-carrying plate. The first ODS of the test TPS panels in all conditions was the torsional mode, which was identical to the first ODS of the load-carrying plate.

The second ODS of the test TPS panels was different for damaged cases. The TPS panel with the damage at the single corner (corner #1) showed an asymmetrical deflection shape which deflected much at the edge close to the damaged corner. Although there was damage in the bolted joints along the diagonal line (corners #1 and #3), the second ODS in this case still did not change much from the healthy state. This might have been due to the symmetry of the damaged positions to the center point. The second ODS of the TPS panel with damage to the bolted joints along the vertical line (corners #1 and 4) had the highest deflection at the two edges (between corners #1 and #4 and between corners #1 and #3) and at the middle areas along the vertical center axis. This bending ODS was symmetric about the vertical center axis.

The third ODS of the test TPS panels also showed different shapes for each damaged case. The bending shape of the TPS panel with the damage at the single corner was slightly different from the healthy case. The third ODS of the TPS panel with damage at two corners along the diagonal line (corners #1 and #3) had a bending shape symmetric to the diagonal line because there was no constraint along the diagonal line due to damages. For the TPS panel with damage at the two corners along the vertical line (corners #1 and #4), the highest deflection was found near the edge (between corners #1 and #4) due to the untied constraint along the vertical line.

Figure 20 shows the MACs between the reference mode shapes of the healthy structure and mode shapes corresponding to the healthy case (Figure 20a) and mode shapes corresponding to the damage cases (Figure 20b−d). The calculated MACs provided information as to how well the shapes correlated. The calculated MAC values of the first four ODSs of the healthy TPS panel were 1 in the main diagonal elements and less than 0.1 in the off-diagonal elements, which meant the MAC calculation was accurate, as shown in Figure 20a. For the single TPS panel presented in this study, the damage states were identified by a threshold of 0.88 (11.6% difference). The calculated MAC between damage and healthy cases indicated significant changes in mode shapes, with values much smaller than 0.88 for all four modes. The most remarkable changes increased with an increasing number of damage corners. A major change occurred in the fourth mode with MAC decreasing to 0.45. Therefore, we confirmed that damage can be detected with MAC reductions.

## 5. Conclusions

We successfully investigated the dynamic characteristics and damage detection using a combination of the 3DPT and MAC methods. The ODSs of the healthy fully-assembled TPS panel or damaged panels were related to the ODSs of the load-carrying plate because of the dominant weight of the load-carrying plate in the metallic TPS panel. The proposed ODS FRF was an effective measurement way to obtain the modal parameters: the natural frequencies, damping ratios, and the ODS of the test structures. The 3DPT provides a fast and accurate way to assess the dynamic characteristics of complex and large structures. In comparison with the conventional method of using accelerometers, the 3DPT demonstrated the advantage of a non-wired setup and simultaneous multi-point measurement data. This advantage reduced the processing time.

In damage detection, a frequency change in a particular natural mode or change in a specific ODS may be a sign of failure. Note that the localized excitation could result in undiscovered modes in the OMA. The use of motion magnification is required for high-frequency modes to detect possible damage to test structures. The obtained ODSs in this study were very useful for identifying unseen damage, such as loosening of the bolted joints in the metallic TPS panel.

Decreased MAC values from case studies showed the damage levels of the metallic TPS panel, specifically, the number of damaged corners. A combination of the 3DPT method and the MAC provided an excellent approach to evaluate the ODSs and to identify the damage states of the metallic TPS panel. We concluded that the proposed method is a practicable non-contact and SHM technique that can be applied with high confidence to study the structural dynamics of a metallic TPS panel and to identify the damage states of a metallic TPS panel due to loose bolted joints.

## Figures and Tables

**Figure 1 sensors-20-07185-f001:**
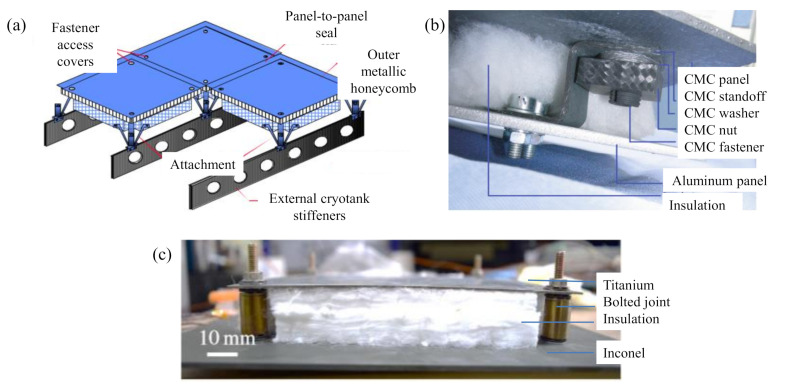
Use of fasteners in thermal protection system (TPS) panels: (**a**) metallic TPS panel designed for the X-33 spaceplane [18], (**b**) ceramic-metallic TPS panel designed for the SHEFEX II [19], and (**c**) metallic TPS panel designed for the current study.

**Figure 2 sensors-20-07185-f002:**
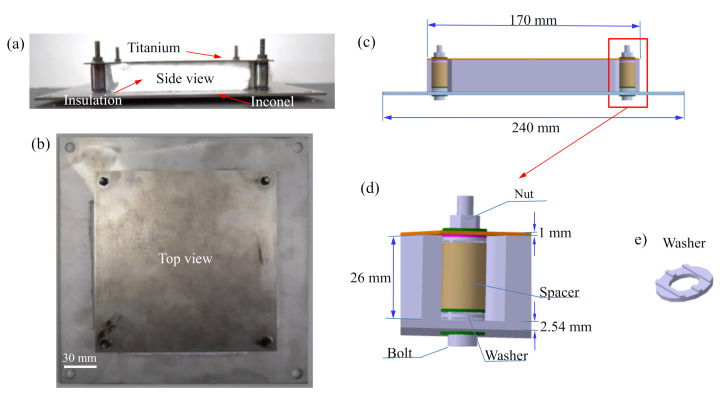
A metallic TPS panel for dynamic testing: (**a**) side view and (**b**) top view of the metallic TPS panel, (**c**) 3D model of the metallic TPS panel, (**d**) the bolted joint, (**e**) washer.

**Figure 3 sensors-20-07185-f003:**
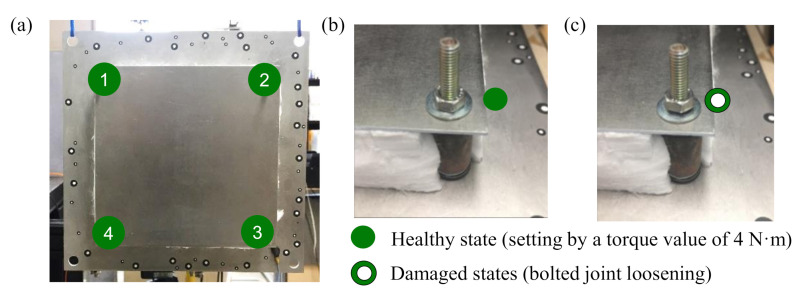
Testing structure. (**a**) the healthy condition of the TPS panel: a torque value of 4 N·m was maintained for four bolted joints at four corners, (**b**) a typical bolted joint in healthy condition, (**c**) a typical case of damaged bolted joint at zero torque value.

**Figure 4 sensors-20-07185-f004:**
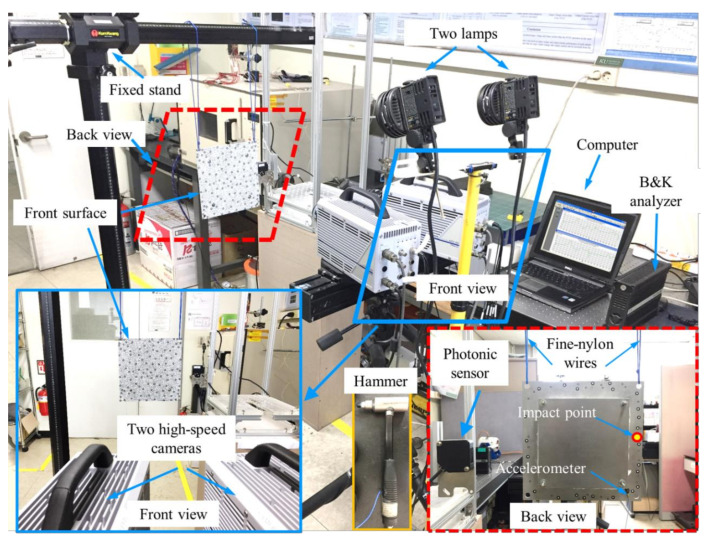
Experimental setup for a dynamic test using high-speed three-dimensional point tracking (3DPT) and an accelerometer.

**Figure 5 sensors-20-07185-f005:**
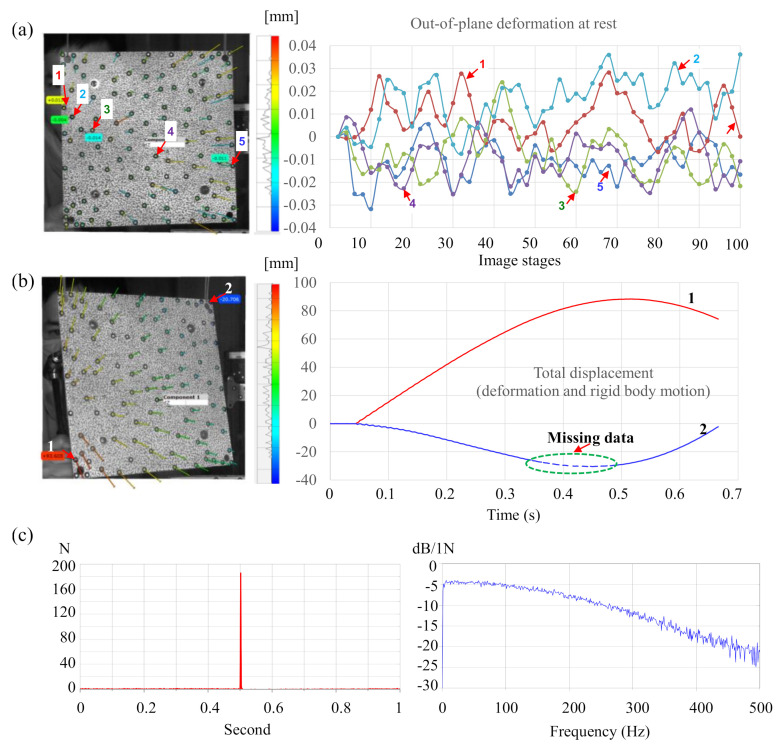
Out-of-plane deformations at rest and displacement at deformed stages: (**a**) out-of-plane deformation at rest, (**b**) out-of-plane displacements of two target points (1 and 2) including rigid body motion. Note: the dashed data of point 2 indicates several missing data, and (**c**) an impulse force history and its autospectrum.

**Figure 6 sensors-20-07185-f006:**
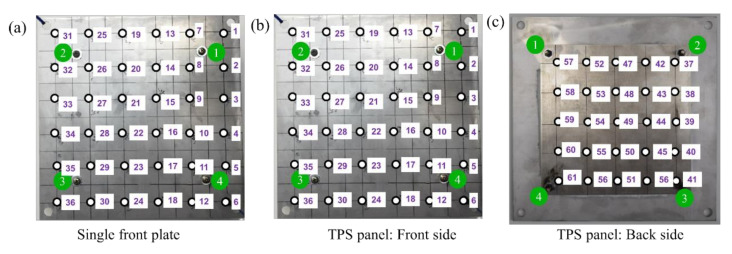
Schematic of test structure and numbering system: (**a**) load-carrying plate with numbering system of 36 marked points, the metallic TPS panel: (**b**) front load-carrying plate and (**c**) back structure (titanium) with numbering system of 61 marked points.

**Figure 7 sensors-20-07185-f007:**
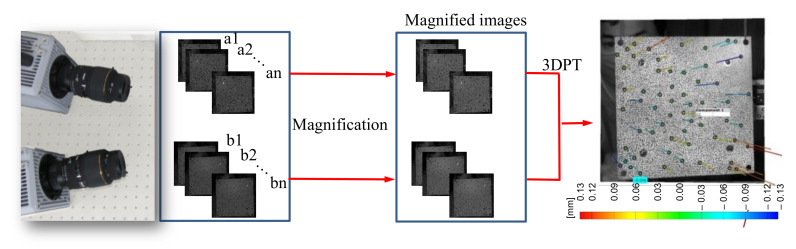
Procedure for capturing a series of images and magnifying captured images using the 3DPT method.

**Figure 8 sensors-20-07185-f008:**
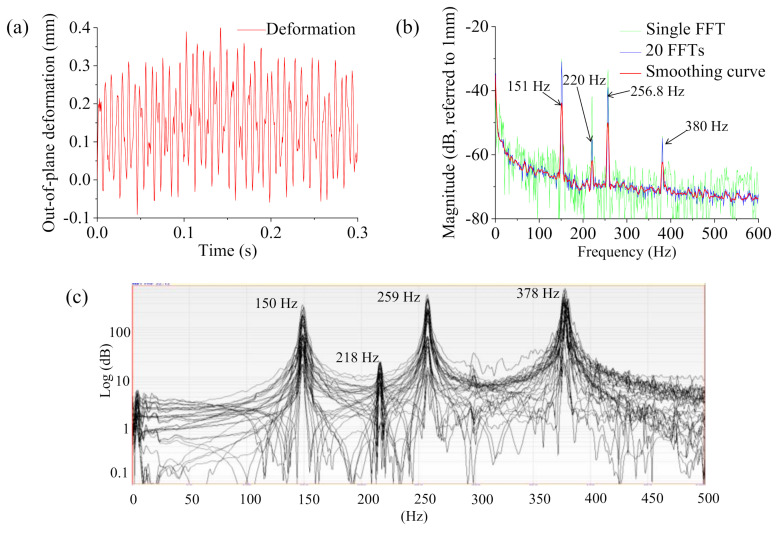
The response and spectral data of the load-carrying plate at a point near the corner: (**a**) time domain, (**b**) frequency domain using 3DPT, (**c**) frequency domain using an accelerometer and roving hammer.

**Figure 9 sensors-20-07185-f009:**
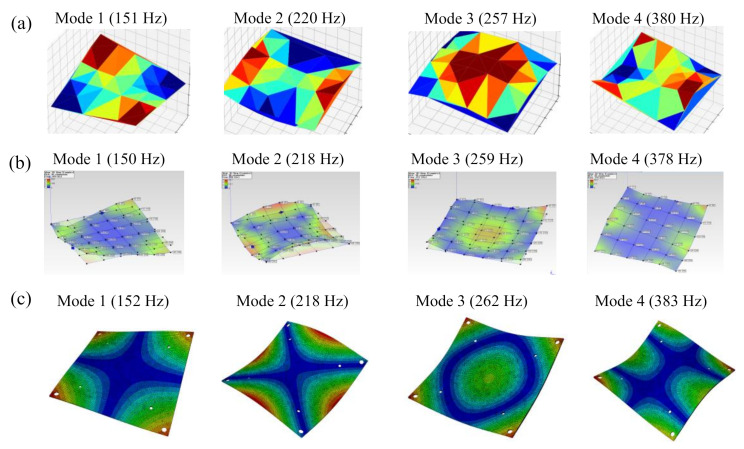
Natural frequencies and mode shapes of the load-carrying plate: (**a**) operational mode shapes with corresponding natural frequencies measured by the 3DPT method, (**b**) mode shapes with corresponding natural frequencies measured by the roving hammer method, (**c**) modal parameters obtained by the finite element analysis (FEA) method.

**Figure 10 sensors-20-07185-f010:**
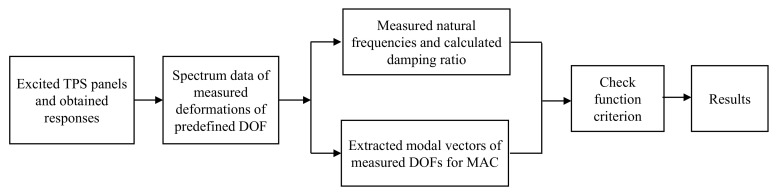
Flowchart of proposed damage detection using 3DPT and modal assurance criterion (MAC) method.

**Figure 11 sensors-20-07185-f011:**
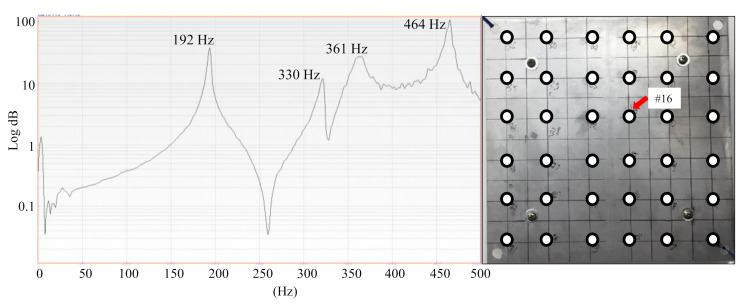
Frequency response function (FRF) of the healthy fully-assembled TPS panel at measurement point 16.

**Figure 12 sensors-20-07185-f012:**
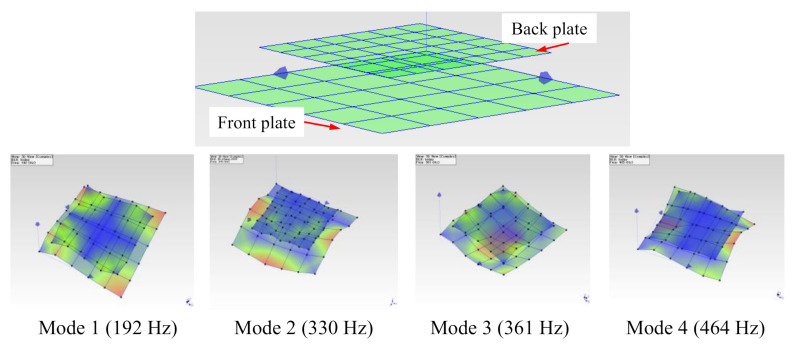
Mode shapes of the healthy fully-assembled TPS panel corresponding to the first four natural frequencies.

**Figure 13 sensors-20-07185-f013:**
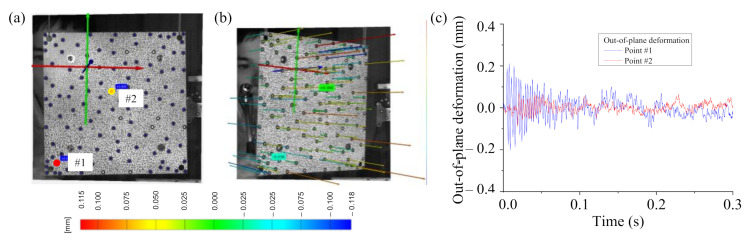
Response of the healthy fully-assembled TPS panel: (**a**) an undeformed state and (**b**) a deformed state with the points of interest, (**c**) response history of the points of interest.

**Figure 14 sensors-20-07185-f014:**
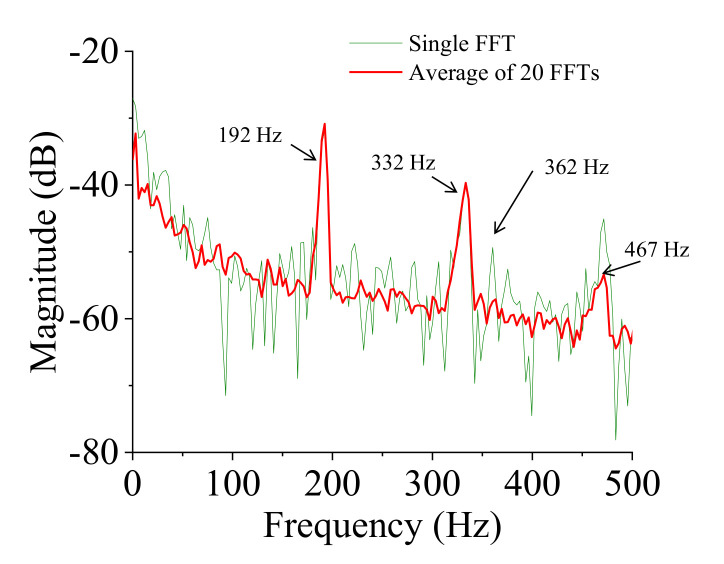
Spectral data of the healthy fully-assembled TPS panel.

**Figure 15 sensors-20-07185-f015:**
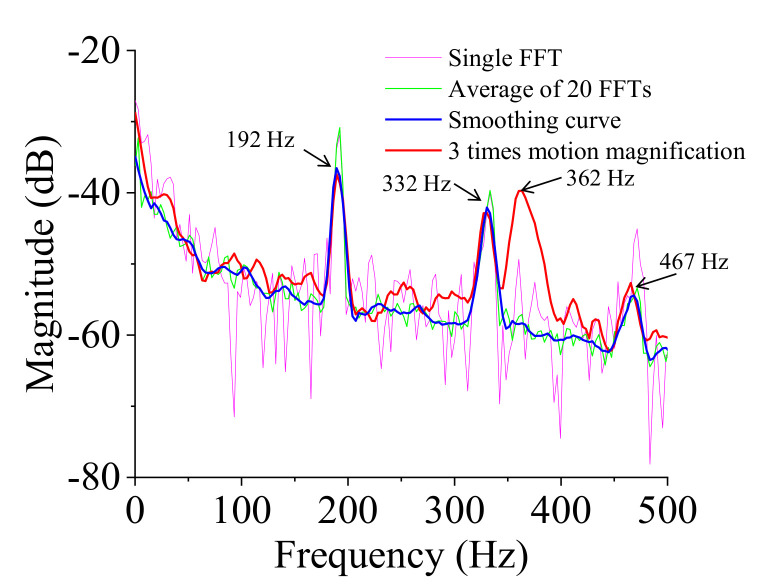
Spectral data of the healthy fully-assembled TPS panel after applying motion magnification.

**Figure 16 sensors-20-07185-f016:**
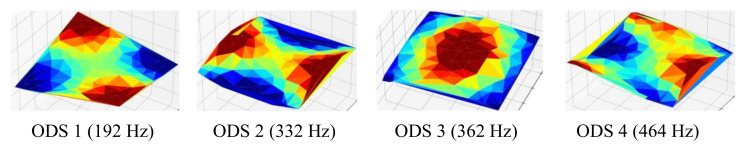
Operating deflection shapes at natural frequencies: healthy fully-assembled TPS panel.

**Figure 17 sensors-20-07185-f017:**
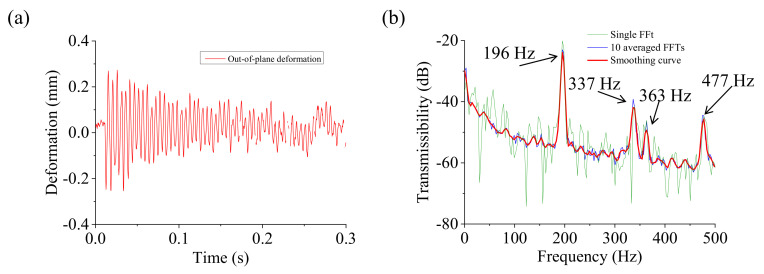
Dynamic characteristics of the skeleton TPS panel (TPS panel without fibrous insulation material): (**a**) deformation response of a point on the front plate of the TPS panel, (**b**) spectral data of the deformation response.

**Figure 18 sensors-20-07185-f018:**
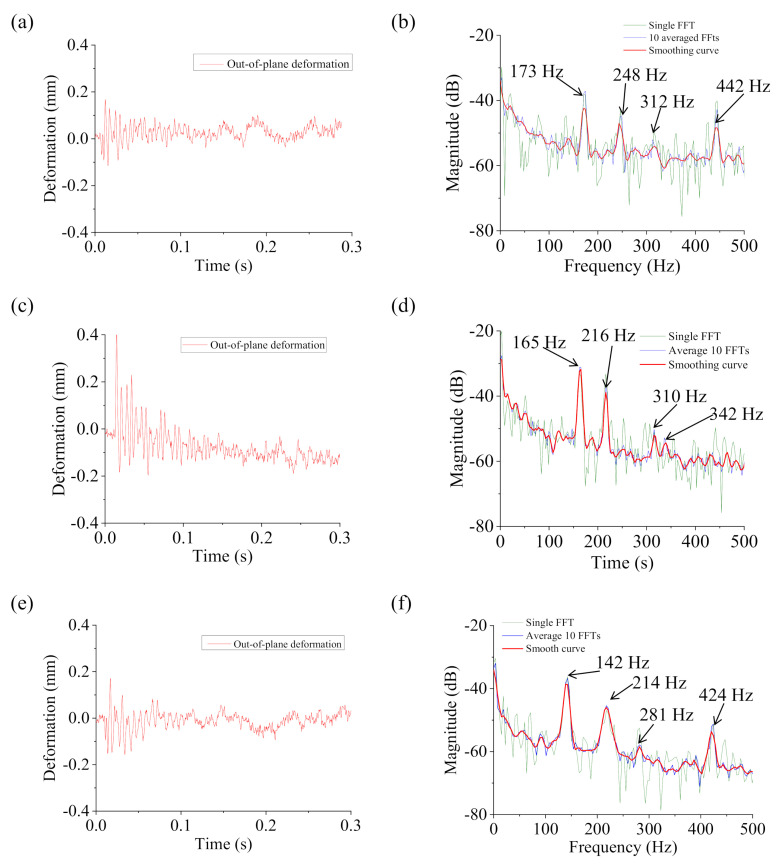
Deformation response and spectral data of the damaged TPS panels: (**a**) deformation response and (**b**) spectral data of damage at corner 1, (**c**) deformation response and (**d**) spectral data of damage at both corners 1 and 3, and (**e**) deformation response and (**f**) spectral data of damage at both corners 1 and 4.

**Figure 19 sensors-20-07185-f019:**
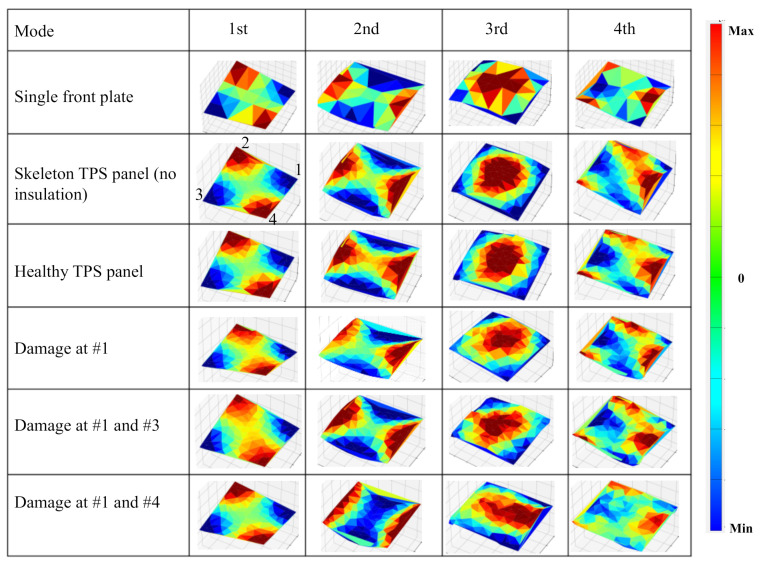
Summary of the operational deflection shapes (ODSs) at the first four natural frequencies: single load-carrying plate, skeleton TPS panel (without fibrous insulation material), healthy assembled TPS panel, damage at specific corners. Note: the order of the corners in all ODSs are the same in the current view.

**Figure 20 sensors-20-07185-f020:**
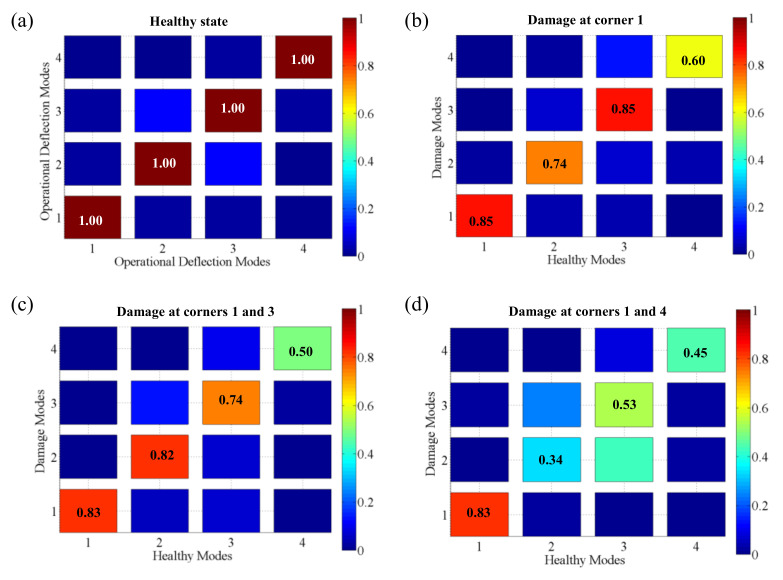
Calculated results of the MAC values: (**a**) self-comparison of a healthy state; comparison between a healthy state and damage states: (**b**) damage at corner #1, (**c**) damage at corners #1 and #3, and (**d**) damage at corners #1 and #4.

**Table 1 sensors-20-07185-t001:** Experimental cases for dynamic characteristics and damage states of the TPS panel.

Components	Load-Carrying Plate	Back Plate	Fibrous Insulation Layer	Fully-Assembled TPS Panel
Size (mm)	240 × 240	170 × 170	170 × 170	
Weight (g)	1229	137	72	1547

**Table 2 sensors-20-07185-t002:** Experimental cases for dynamic characteristics and damage states of the TPS panel.

Case	Load-carrying Plate	Assembled TPS Panel without Fibrous Insulation Material	Assembled TPS Panel with Fibrous Insulation Material			
State	Single plate	Healthy condition	Healthy condition	Damage at specific corners(#)		
				#1	#1 and #3	#1 and #4

**Table 3 sensors-20-07185-t003:** Summary of the change in natural frequencies of the test cases.

Mode	Load-Carrying Plate (Hz)	Assembled Panel with Insulation (Hz)	Assembled Panel without Insulation (Hz)	Damage at Specific Corners (#)					
				#1		#1 and #3		#1 and #4	
				(Hz)	% diff.	(Hz)	% diff.	(Hz)	% diff.
1	151	192	196	173	9.9	165	14	142	26
2	220	332	337	248	25.3	216	34.9	214	35.5
3	256	362	363	312	13.8	310	14.3	281	22.3
4	380	467	477	442	5.3	342	26.7	424	9.2

**Table 4 sensors-20-07185-t004:** Summary of the damping ratio of the first natural mode.

Mode	Load-Carrying Plate	Assembled Panel with Insulation	Assembled Panel without Insulation	Damage at Corner #1	Damage at Corners #1 and #3	Damage at Corners #1 and #4
1	0.9%	1.1%	1.3%	1.63%	1.7%	2.4%

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
