# Peer review of "Dynamic Characteristics and Damage Detection of a Metallic Thermal Protection System Panel Using a Three-Dimensional Point Tracking Method and a Modal Assurance Criterion"

_sensors, 2020, doi:10.3390/s20247185_

Round 1

Reviewer 1 Report

I would like to congratulate you on the quality of your paper. Therefore, I approve it for publication asking only that you pass once more on the structural correction of the text.

Author Response

Thank you very much. We have checked and corrected several words and sentences throughout the manuscript.

Reviewer 2 Report

The paper proposes a novel method for the dynamic characterisation and vibration-based damage assessment of a panel. The panel is intended for the thermal protection system (TPS) of a hypersonic vehicle. In this regard, the paper deals mainly with the issue of fastener failures, which is a common fault for this kind of structural components. The aim is of great interest and very timely for aerospace engineers and researchers in structural dynamics.

The proposed method is based on OMA, which is an output-only approach of great practical usefulness, and on video-based, non-contact measurements.  

The study is interesting and worthy of publication, yet there are several issues, both in the content and in the editing and writing, that must be solved before acceptance. Especially the form, including the figures, must be improved. Specifically:

  1. Section 2.1: the whole discussion about the eigenproblem is well-known for any researcher in the field of structural dynamics and can be avoided, also because it is not very precise (mode shapes and eigenvectors are related by a simple transformation but they are not exactly synonymous; the poles are complex conjugate pairs only for oscillating systems – non-oscillatory decay has no practical interest in vibration-based SHM but it may occur nevertheless; the real part is not the modal damping but proportional to it; and the imaginary part is not the modal frequency but proportional to the damped angular frequency)
  2. In Section 2.2, the steps leading to the definition of the FRF can be omitted. Equation 6 is not needed as long as one can refer to Equation 4. Figure 2 explains well the difference between roving input channel and roving output channel, but since neither of these two techniques is the main aim of the proposed method (which relies on simultaneous multi-response measurement, i.e., a SIMO identification) it could be omitted.
  3. In general, the paper is quite long. The authors should try to synthesise their discussion, especially in the first Sections, as many theoretical aspects are assumed to be well-known to the intended audience.
  4. Section 2, the authors should indicate that eq (1) is the governing equation of motion of a Multi-DoFs system.
  5. Section 3.3: it is not clear what the authors mean with “Savitzky-Golay smoothing method”. Do they refer to a Savitzky-Golay sliding polynomial filter? If so, which order and window width have been used?
  6. Finite Element Model (Section 3.3): more details about the FEA are needed. Which commercial software has been used? Ansys Mechanical? Which elements (and how many) have been used?
  7. Video processing is gaining more and more attention for Structural Health Monitoring and the dynamic characterisation of aerospace components. Some recent researches were published here on Sensors as well; for instance, the target tracking was applied in https://doi.org/10.3390/s19102345 for the estimation of the nonlinear behaviour of a prototype wing undergoing large oscillations, avoiding to load the flexible structure with the sensors’ masses. Other techniques were reviewed in https://doi.org/10.1088/1742-6596/1249/1/012004. The phase-based motion magnification (PBMM) has also been applied for damage detection, severity assessment, and localisation, such as e.g. in https://doi.org/10.1111/str.12336. The authors may consider adding these references to their manuscript.
  8. Section 4: It is not clear what the ScopeTM software is (is it the same ME’scope software from Vibrant technology, Inc. mentioned before?). From the context, it seems the software utilised to define the mode shapes reported in the second row of Figure 10 and Figure 13. What input data were utilised? The benchmark data from the physically-attached sensors? If the data from the video acquisition were utilised, it is not clear what is the difference between the first and the second row of Figure 10. 

Some minor/formal issues include:

  1. Please carefully review your English. The text is overall good, but there are several mistakes and typos throughout the whole manuscript (e.g. page 5, The solution to Eq. (4) *are the* ODSs. ).
  2. The explicit subdivision of the abstract in (1) background, (2) methods, and (3) results, while not incorrect, is a bit unusual. The authors may consider omitting the numbers in the bracket and the subtitles.
  3. Abstract, lines 11-12: to protect *a* hypersonic vehicle.
  4. Page 4, line 146: “We focus on detecting global damage rather than localized damage”. Actually, a loose bolt is localised damage. Its effects on the dynamic response are global but the damage is still limited in a well-defined location.
  5. Figure  1 is well made but: (1) the text font (both in (a) and (b)) is too small to be easily readable. (2) it is not clear why the authors decided to retrieve some images from the literature rather than showing a picture of their actual experimental case study (which is reported in Figures 3 and 4).
  6. Page 3, line 98, the sentence “which are able to measure displacements or temperature without invading the structures” is not supported by any reference. The authors may add some examples, especially for the non-contact measurement of temperature. Can it be performed by means of computer vision as well? or this sentence refers to standard thermal camera approaches mentioned later?
  7. Page 3, line 101, “the non-contact method using a laser has difficulty with large displacement measurement”. Actually, in case of large rigid motions such as e.g. large flapwise deflections, laser measurements are useless without laser tracking or corrections in post-processing (and both requires some knowledge about the rigid motion itself, which is often unavailable a priori).
  8. Page 4, lines 140- 141: “a * variant * of the non-contact image-based method” it may be more correct to say just  “non-contact image-based method”.
  9. Page 4, line 148, the acronym ODS is presented without being previously defined (it is explained but only in the abstract and not in the text)
  10. Section 3: the whole introduction “This section may be divided by subheadings. It should provide a concise and precise description of the experimental results, their interpretation as well as the experimental conclusions that can be drawn.” Is implicit and should be omitted.
  11. Figure 4, the torque value should be in ; it seems more like  as it is now.
  12. Figure  6: (1) the text font is too small to be easily readable. (2) in (a), it is not clear to which points the time series refer to, and the x-axis label is missing. (3) in (b), point 2 is indicated on blue in the left figure and black in the right side plot. It is not clear why a portion of the black curve is dashed. The x-axis label is missing as well. (4) in (c), on the left side, the picture of the impulse is not very helpful as it is now, and the plot on the right side should be in a colour easier to notice on the white background (light grey is not adapt).
  13. Figure 7: the labels with the numbers of the 36 (in (b)) and 61 (in (d)) points are small and difficult to read.
  14. Figure 8: the colour bar is too small to be readable. Same for the time history
  15. Figure 9: (1) the plot on the (a) should be in a colour easier to notice on the white background (light grey is not adapt). (2) the font size in the x- and y-axes of (c) is too small to be readable
  16. Figure 10: for the last row, it could be better to crop out the mode shapes from the screenshot, not reporting the rest of the window (which is of no interest and again too small to be readable).
  17. Figures 18 and 19: the plots should be in a colour easier to notice on the white background (light grey is not adapt).
  18. 3DPT-based ARAMIS® software: it may be useful to add the link to the software, as a reference or a footnote.

Author Response

We would like to thank the reviewer for your interest in our work and for helpful comments that will improve the manuscript greatly. We have checked all the general and specific comments provided and have made necessary changes accordingly. Please see below for your evaluation.

Genneral comments

The paper proposes a novel method for the dynamic characterisation and vibration-based damage assessment of a panel. The panel is intended for the thermal protection system (TPS) of a hypersonic vehicle. In this regard, the paper deals mainly with the issue of fastener failures, which is a common fault for this kind of structural components. The aim is of great interest and very timely for aerospace engineers and researchers in structural dynamics.

The proposed method is based on OMA, which is an output-only approach of great practical usefulness, and on video-based, non-contact measurements.  

The study is interesting and worthy of publication, yet there are several issues, both in the content and in the editing and writing, that must be solved before acceptance. Especially the form, including the figures, must be improved. Specifically:

Thank you very much for your kind reviews. We have tried to improve the format of our manuscript as much as possible.

Major comments

1. Section 2.1: the whole discussion about the eigenproblem is well-known for any researcher in the field of structural dynamics and can be avoided, also because it is not very precise (mode shapes and eigenvectors are related by a simple transformation but they are not exactly synonymous; the poles are complex conjugate pairs only for oscillating systems – non-oscillatory decay has no practical interest in vibration-based SHM but it may occur nevertheless; the real part is not the modal damping but proportional to it; and the imaginary part is not the modal frequency but proportional to the damped angular frequency)

Thank you. We shortened Section 2. We re-wrote Section 2 with a reduction of the discussion of eigenproblems in the structural vibration.

2. In Section 2.2, the steps leading to the definition of the FRF can be omitted. Equation 6 is not needed as long as one can refer to Equation 4. Figure 2 explains well the difference between roving input channel and roving output channel, but since neither of these two techniques is the main aim of the proposed method (which relies on simultaneous multi-response measurement, i.e., a SIMO identification) it could be omitted.

Thank you very much. We have deleted Fig. 2 in the revised version and the Figures in the revision are reordered.

3. In general, the paper is quite long. The authors should try to synthesise their discussion, especially in the first Sections, as many theoretical aspects are assumed to be well-known to the intended audience.

Thank you very much. We tried to shorten discussions in Section 2.

4. Section 2, the authors should indicate that eq (1) is the governing equation of motion of a Multi-DoFs system.

Thank you. We corrected it as the governing equation of a multi-DOF system.

5. Section 3.3: it is not clear what the authors mean with “Savitzky-Golay smoothing method”. Do they refer to a Savitzky-Golay sliding polynomial filter? If so, which order and window width have been used?

Yes, we used the Savitzky-Golay smoothing filter with an order of 3 and a window width of 5. We would like to add those in the revised version.

6. Finite Element Model (Section 3.3): more details about the FEA are needed. Which commercial software has been used? Ansys Mechanical? Which elements (and how many) have been used?

We used ABAQUS software to simulate the frequency analysis of the front loading-carrying plate. C3D8R elements with an 8-node linear brick and reduced integration were used in the simulation. There were 78148 elements in total. We would like to add details about FEM model to the manuscript.

7. Video processing is gaining more and more attention for Structural Health Monitoring and the dynamic characterisation of aerospace components. Some recent researches were published here on Sensors as well; for instance, the target tracking was applied in 10.3390/s19102345 for the estimation of the nonlinear behaviour of a prototype wing undergoing large oscillations, avoiding to load the flexible structure with the sensors’ masses. Other techniques were reviewed in 10.1088/1742-6596/1249/1/012004. The phase-based motion magnification (PBMM) has also been applied for damage detection, severity assessment, and localisation, such as e.g. in 10.1111/str.12336. The authors may consider adding these references to their manuscript.

Thank you very much for your helpful recommendations. We considered carefully these references and added some references in the revised version of the manuscript. 

8. Section 4: It is not clear what the ScopeTM software is (is it the same ME’scope software from Vibrant technology, Inc. mentioned before?). From the context, it seems the software utilised to define the mode shapes reported in the second row of Figure 10 and Figure 13. What input data were utilised? The benchmark data from the physically-attached sensors? If the data from the video acquisition were utilised, it is not clear what is the difference between the first and the second row of Figure 10. 

Yes, it is the ME’scopeTM software (Vibrant technology, Inc.). Those inconsistent words are our typos. We would like to correct them.

We used the ME’scopeTM software to define the mode shapes in the Figs. 9b and 12 (previous version: Figs.10b and 13). As we mentioned in the manuscript, we used two experimental methods, one was the roving hammer (hammer and accelerometer) and another one was the multi-response measurement (image-based 3DPT).

To obtain the mode shapes in Figs. 9b and 12, we used the output data from the physically-attached accelerometer and the input data from the impact hammer. To obtain the mode shapes in Fig. 10a, the output data from the video acquisition was only used for extracting the mode shapes by an in-house python code in the ARAMIS® software.

In summary, we used ME’scopeTM software to define the mode shapes of the experimental modal analysis (EMA) while the ARAMIS® software was used to define the mode shapes of the operational model analysis (OMA).  

 Minor comments

1. Please carefully review your English. The text is overall good, but there are several mistakes and typos throughout the whole manuscript (e.g. page 5, The solution to Eq. (4) *are the* ODSs. ).

Thank you. We tried to correct several typos and mistakes throughout the manuscript.

2. The explicit subdivision of the abstract in (1) background, (2) methods, and (3) results, while not incorrect, is a bit unusual. The authors may consider omitting the numbers in the bracket and the subtitles.

Thank you. We deleted the subtitles.

3. Abstract, lines 11-12: to protect *a* hypersonic vehicle.

Thank you. We corrected it.

4. Page 4, line 146: “We focus on detecting global damage rather than localized damage”. Actually, a loose bolt is localised damage. Its effects on the dynamic response are global but the damage is still limited in a well-defined location.

Thank you for your comment. We agreed with the reviewer that a loose bolt is at a specific local position which can be considered as localized damage. However, we only focus on detecting damage existence rather than localizing the damage position exactly. Thus, we investigated changes in modal parameters with different numbers of damaged corners corresponding to three different damage levels.

We would like to correct that sentence in the revised version.

5. Figure 1 is well made but: (1) the text font (both in (a) and (b)) is too small to be easily readable. (2) it is not clear why the authors decided to retrieve some images from the literature rather than showing a picture of their actual experimental case study (which is reported in Figures 3 and 4).

Thank you very much. (1) We enlarged the font size of the text in the Fig. 1. (2) We aimed to show some recent TPS panels that used fasteners as structural connections. We added one more image in Fig. 1 to show our present TPS panel.

6. Page 3, line 98, the sentence “which are able to measure displacements or temperature without invading the structures” is not supported by any reference. The authors may add some examples, especially for the non-contact measurement of temperature. Can it be performed by means of computer vision as well? or this sentence refers to standard thermal camera approaches mentioned later?

Thank you for your dedicated reviews. We added several references to support the above mentioned sentence.

7. Page 3, line 101, “the non-contact method using a laser has difficulty with large displacement measurement”. Actually, in case of large rigid motions such as e.g. large flapwise deflections, laser measurements are useless without laser tracking or corrections in post-processing (and both requires some knowledge about the rigid motion itself, which is often unavailable a priori).

Yes, we meant that the non-contact method using a laser has difficulty with a large displacement measurement and also with the presence of large rigid motions. In addition, displacement laser sensor often has a limit range of displacements. We added some expalnations to the manuscript.

8. Page 4, lines 140- 141: “a * variant * of the non-contact image-based method” it may be more correct to say just  “non-contact image-based method”.

Thank you. We corrected them.

9. Page 4, line 148, the acronym ODS is presented without being previously defined (it is explained but only in the abstract and not in the text)

Thank you. We added the definition of the acronym ODS at line 113 in page 3.

10. Section 3: the whole introduction “This section may be divided by subheadings. It should provide a concise and precise description of the experimental results, their interpretation as well as the experimental conclusions that can be drawn.” Is implicit and should be omitted.

Thank you. We deleted them.

11. Figure 4, the torque value should be in ; it seems more like  as it is now.

We used a torque value of 4 N·m as indicated in Fig. 3 (previous version: Fig. 4)

12. Figure  6: (1) the text font is too small to be easily readable. (2) in (a), it is not clear to which points the time series refer to, and the x-axis label is missing. (3) in (b), point 2 is indicated on blue in the left figure and black in the right side plot. It is not clear why a portion of the black curve is dashed. The x-axis label is missing as well. (4) in (c), on the left side, the picture of the impulse is not very helpful as it is now, and the plot on the right side should be in a colour easier to notice on the white background (light grey is not adapt).

Thank you. (1) We enlarged the font size of the text in Fig. 5 (previous version: Fig. 6). (2) We added the point code and the x-axis label. (3) We corrected color code for the two curves. A portion of the dashed data in curve 2 means that there are several missing data. (4) We aimed to show the applied impulse load that impacted the TPS panel to create a displacement not exceeding the measurement volume (less than 100 mm in this study). We also changed color of the curve.   

13. Figure 7: the labels with the numbers of the 36 (in (b)) and 61 (in (d)) points are small and difficult to read.

Thank you. We enlarged the font size of the labels in Fig. 6 (previous version: Fig. 7).

14. Figure 8: the colour bar is too small to be readable. Same for the time history

Thank you. We enlarged the font size of the color bar’s scale in Fig. 7 (previous version: Fig. 8).

15. Figure 9: (1) the plot on the (a) should be in a colour easier to notice on the white background (light grey is not adapt). (2) the font size in the x- and y-axes of (c) is too small to be readable

Thank you. We changed the color of the curve and enlarged the font size of the x- and y-axes in Fig. 8 (previous version: Fig. 9).

16. Figure 10: for the last row, it could be better to crop out the mode shapes from the screenshot, not reporting the rest of the window (which is of no interest and again too small to be readable).

Thank you. We cropped out the pictures in Fig. 9c (previous version: Fig. 10c).

17. Figures 18 and 19: the plots should be in a colour easier to notice on the white background (light grey is not adapt).

Thank you. We changed the color of the curves in Figs. 17 and 18 (previous version: Figs. 18 and 19).

18. 3DPT-based ARAMIS® software: it may be useful to add the link to the software, as a reference or a footnote.

Thank you. We provided a reference to find the information of the 3DPT-based ARAMIS® software.

Reviewer 3 Report

This article is well written and technically sound. I recommend publication of the manuscript. 

Author Response

Thank you very much for your review.

Reviewer 4 Report

Methods of damage detection of structures using modal analysis are well known from many years. The method proposed by Authors is not new in my opinion. Authors should clearly stressed what is original in this method taking into account literature of the subject.

Authors didn't shown any advantages of their method in comparison with methods well known form the literature.

I have not found any information about sensitivity of the proposed method. From pratical point of view such information is crucial for the damage detection systems.

Also literature cited in the paper is unappropraite. Many very important papers concerned damage detection methods by modal analysis were omitted.

I propose reconsider the paper after major revison.

Author Response

Thank you for your rigorous review and all of your comments.

We agreed with the reviewer that damage detection using modal analysis is not a new method. So far people have used the conventional method to detect possible damage in structures by considering changes in modal parameters. These were the contact and pointwise methods (accelerometers and laser vibrometers) or in-plane displacement measurement method (image-based methods). In this study, we present a new approach method based on the non-contact optical technique that takes advantage of three-dimensional (3D), non-contact, large-scale, and multi-data measurements [60].

As far as we know that the 3D digital image correlation (DIC) is the optical, full-field, and non-contact measurement technique. Thus, the DIC technique becomes popular in the modal analysis [56-58]. However, the DIC technique has several disadvantages in measuring a large-scale structure such as an aerospace structure. Because of the full-field data measurement, there would be a large data storage and much computational time in the DIC method. In addition, making a speckle pattern for the DIC measurement would take more time than attaching discrete points in the presented three-dimensional point tracking (3DPT) measurement technique. Therefore, we proposed the 3DPT measurement technique to investigate the dynamic characteristics and damage existence of a metallic thermal protection system (TPS) panel. By taking the advantage of multi-predefined points measurement, the 3DPT can replace multi-laser vibrometer sensors with the simultaneous measurement and also can replace the DIC method with the shorter processing time.

Within the scope of the paper, any signs of damage that cause changes in modal parameters are important in the TPS panel. When a sign of the damage was detected, the TPS panel would be inspected immediately by removing the TPS panel from the vehicle's structure. Thus, we only focus on damage existence in the TPS panel rather than localizing damage positions exactly. Three damage levels represented three cases of the number of damaged corners. Therefore, the use of modal assurance criterion (MAC) is suitable for detecting changes in general mode shapes.

We have obtained a threshold value for the damage detection in a single TPS panel. This represented a maximum measurement uncertainty of the proposed method. It was about 11.6%. We added in the revision.

In our previous study [10], we investigated the thermomechanical performance of the metallic TPS panel in several thermal cycles. We found that there was a permanent deformation that was due to plastic deformation of the bolted joints. These permanent deformations would result in the reduction of the clamping force of the bolted joints. So, the TPS panel would be capable of being wounded to deflection and damage under vibration loads in the next flight mission. Therefore, the damage due to bolted joint failure is the main concern in our current research.

We tried to include many references that used modal analysis for damage detection. We might not cover all the papers. If the reviewer thinks that any important papers are missing and need to be included, please let us know. We will appreciate your suggestions. We have revised our manuscript as much as possible.

Reference

  1. P. Poozesh, J. Baqersad, C. Niezrecki, P. Avitabile, E. Harvey, R. Yarala. "Large-area photogrammetry based testing of wind turbine blades," Mechanical Systems and Signal Processing Vol. 86, 2017, pp. 98-115, doi: 10.1016/j.ymssp.2016.07.021
  2. Le, V.T.; Goo, N.S.; Kim, J.Y. Thermomechanical behavior of superalloy thermal protection system under aerodynamic heating. Journal of Spacecraft and Rockets 2019, 56, 1432-1448, doi:10.2514/1.a34400.
  3. Srivastava, V.; Baqersad, J. An optical-based technique to obtain operating deflection shapes of structures with complex geometries. Mechanical Systems and Signal Processing 2019, 128, 69-81, doi:10.1016/j.ymssp.2019.03.021.
  4. Beberniss, T.J.; Ehrhardt, D.A. High-speed 3D digital image correlation vibration measurement: Recent advancements and noted limitations. Mechanical Systems and Signal Processing 2017, 86, 35-48, doi:10.1016/j.ymssp.2016.04.014.
  5. Wu, R.; Zhang, D.; Yu, Q.; Jiang, Y.; Arola, D. Health monitoring of wind turbine blades in operation using three-dimensional digital image correlation. Mechanical Systems and Signal Processing 2019, 130, 470-483, doi:10.1016/j.ymssp.2019.05.031.
  6. Poozesh, P.; Baqersad, J.; Niezrecki, C.; Avitabile, P.; Harvey, E.; Yarala, R. Large-area photogrammetry based testing of wind turbine blades. Mechanical Systems and Signal Processing 2017, 86, 98-115, doi:10.1016/j.ymssp.2016.07.021.

New version in Introduction:

A thermal protection system (TPS) is necessary for hypersonic aircraft and spacecraft to withstand aerodynamic heating and acoustic loads during hypersonic flight. The TPS panel serves as a shielding cover for the fuselage of the vehicle and endures possible impact damage from debris [1,2]. Several types of TPS, such as metallic TPS [3], multilayer TPS [4], integrated TPS [5], and bioinspired TPS [6], have been proposed and investigated through the thermal [7-9] and thermo-mechanical performance [10] tests.

A vehicle’s fuselage can be exposed to more extreme thermal, pressure, and impact loads if a TPS panel becomes damaged [11-14]. Thus, the TPS panel must be in good condition before launch because of its critical role in protecting the vehicle’s structures, subsystems, and even humans. Understanding the structural response and damage state of the TPS panel is important in design and maintenance processes. Chen et al. [15] presented an analysis method to study the panel flutter of a metallic TPS. They calculated natural frequencies and predicted critical dynamic flutter. They concluded that the outer sandwich of the metallic TPS panel was not susceptible to panel flutter but the panel-to-panel seals of the metallic TPS panel were susceptible to panel flutter. Tobe et al. [16] developed a method for localizing fastener failure and material optimization in TPS panels. They employed accelerometers to measure dynamic response of TPS panels under an impulse force. However, due to restrictions of the experimental apparatus, the dynamic characteristics of the TPS panel could not be sufficiently considered and their damage investigation of the TPS panel was limited. Since the inspection of the TPS panel by traditional methods is laborious and expensive [17], what remains to be developed is a new and simple inspection method that has high accuracy and less inspection time. The purpose of this paper is to present a structural health monitoring (SHM) method which improves the performance of conventional methods for investigating the damage state of the TPS panel in the laboratory.

We designed metallic TPS panels to be connected to the vehicle’s fuselage via mechanically bolted joints that provide functions of structural connections, energy dissipation, and vibration damping in a metallic TPS panel. Figs. 1 a and b show an example of TPS panels with fasteners as structural connections at four corners of the panel. Damage modes of bolted joints such as self-loosening, fatigue, and separation can lead to the penetration of hot air into the interior of the vehicle, which might result in the catastrophic loss of the vehicle. In our previous study [10], a metallic TPS panel was investigated in a thermomechanical experiment and simulation. Fig. 1c shows a picture of the current metallic TPS panel. We found that after several simulated missions, there was a permanent deformation of the metallic TPS panel that was due to the plastic deformation of the bolted joint structures, which resulted in the reduction of the clamping forces of the fasteners. Consequently, the TPS panel would be vulnerable to deflection and damage under vibration loads in the next mission, and thus the goal of TPS reusability would not be achieved. Therefore, the damage due to bolted joint failure is the main concern in our current research.

Fig. 1. Use of fasteners in TPS panels: (a) metallic TPS panel designed for the X-33 spaceplane [18], (b) ceramic-metallic TPS panel designed for the SHEFEX II [19], and (c) metallic TPS panel designed for the current study.

One of the traditional methods to detect damage employs visual inspection. This method is adequate if the damage is visible on the surface of the TPS panel, such as cracks, holes and fractures. However, if the features are not visible on the surface, such as damage to a bolted joint or damage in interior components, then visual inspection is not sufficient. The primary methods used to detect bolted joint damage or invisible damage usually utilize the dynamic characteristics of the structure. Damage to structures causes changes to the modal parameters of the structure, such as the natural frequency, damping ratio, and mode shape, compared with the initial or fully healthy state. Therefore, bolted joint damage in structures also causes changes in the dynamic characteristics [16]. Conversely, we can detect the damage or failure by identifying the changes in the modal parameters, such as the natural frequencies [20] and the damping ratios [21].

Along with changes in natural frequencies and damping ratios, a change in mode shapes is also considered for damage detections. The mode shapes are less sensitive to environmental effects, such as temperature, than natural frequencies [22]. The modal assurance criterion (MAC), which indicates changes in mode shapes, have commonly been used for model validation [23]. Orlowitz et al. [24] used accelerometers mounted on a square-plate structure to measure the structural dynamics. A large number of predefined points on the structure were used to obtain mode shapes, and MAC values were compared to each other to validate the proposed experimental methods. Nguyen et al. [25] used a laser vibrometer to obtain mode shapes of a composite disc from a large number of predefined measurement points and calculated MACs for damage detection. The use of the MAC in damage detection requires detailed mode shapes (many points) and higher frequency modes to obtain better results [26]. Other considerations of damage detections are the Modal Curvature method (MCM) [27] and Modal Flexibility method (MFM) [28]. Pandey et al. [29] presented a damage detection method based on changes in mode shape curvature which was known as the MCM. The curvature values were computed from the displacement mode shape using the central difference operator. It assumed that damage-associated stiffness reduction increases the curvature. The damage localization can be determined by evaluating the largest computed MCM value. This methodology demonstrated a high level of damage sensitivity. However, the MCM also presented some drawbacks; one of them is errors due to the application of the central difference approximation method to displacement mode shapes [30]. It might result in a false damage localization [31]. The MCM alone is not recommended for damage identification, it may be used in conjunction with other sub-optimal modal parameters [32]. Later on Pandey and Biswas [28] proposed an approach for damage detection based on flexibility change of the structure. The MFM defines the flexibility matrix as the inverse of the stiffness matrix. The flexibility matrix could be determined with fewer modes than was required for the stiffness matrix. Estimation of damage locations based on the modal flexibility method depends on the number of sensors used and distance of the sensors to the damaged location.

Several experimental methods have been used for damage detection of structures: accelerometers [16,33], laser displacement measurement [34,35], photogrammetry [36-39], infrared thermography [40], and ultrasound [41]. Among them, the method that uses accelerometers has the effect of mass loading for lightweight structures and a labor-intensive and time-consuming process for large structures. In addition, the contact measurement is pointwise and limited to only a few locations. These limitations have been solved by recent progress in non-contact measurement methods, which are able to measure displacements or temperature without invading the structures [34,40,42]. In the non-contact method using infrared thermography, the structure being heated detects cracks or damage, while the non-contact method using ultrasound is also suitable for cracks or damage in small-scale structures. Moreover, the non-contact method using a laser has difficulty with large displacement measurement or the presence of large rigid body motions and needs much acquisition time for a large number of measurement points. In addition, the full-field mode shape measurement requires much inspection time in these existing contact and non-contact methods. 

The non-contact method using photogrammetry along with vibration tests is preferred for damage detection based on the changes in the dynamic characteristics of structures [43]. By tracking movements of exterior features of a structure that appear on digital images captured by cameras, the displacement can be calculated by photogrammetry theory and its modal parameters can be estimated for damage detection. The digital image correlation (DIC) method is a kind of photogrammetry method that measures the full-field deformation of structure in various fields of research, such as for vibration [44], crack monitoring [45-47], high-temperature structures [48-50], electronic packaging [51-53], and small structures [54,55]. The identification of a whole area displacement field or operating deflection shape (ODS) is one of the most interesting features using the DIC method [56]. Thus, the use of the DIC method in vibration and modal analysis has become common [57]. Wu et al. [58] used the DIC method to measure displacements and strains in rotating wind turbine blades, and they used the blade displacement in the frequency domain to identify faults. Helfrick et al. [59] used the DIC to detect damage based on changes in curvature of the structure’s displaced shape.

Moreover, the three-dimensional point tracking (3DPT) method is also a kind of photogrammetry method that uses a pair of digital cameras to measure 3D displacements of discrete points attached to a structure [60]. The 3DPT uses removable optical targets attached to the surface of a structure. The removable optical targets make the 3DPT method preferable if the structures in service are inspected and require decontamination to the test structure. A TPS panel assembled on a hypersonic vehicle is an example. The use of the 3DPT method in vibration analysis has become very popular, and both laboratory vibration measurements and large-scale outdoor measurements have been reported [60-62]. Poozesh et al. [60] successfully tested and validated the 3DPT method in a large-area measurement of wind turbine blades. Warren et al. [61] compared the mode shapes of a base-upright structure obtained from accelerometers, 3D laser vibrometers, full-field DIC, and 3DPT methods. MAC values were determined to validate the measurement methods. They reviewed similar results from each method and showed that the 3DPT method was the most suitable method for measuring the dynamic characteristics of a structure. In addition, for a typical application that requires a small amount of data storage and short computation time, the 3DPT is preferable over the DIC method because the measurement of the discrete points has a smaller amount of data storage [61]. By attaching multiple points on a testing structure, the 3DPT becomes a special laser vibrometer measurement method that can simultaneously measure responses of predefined points. Note that any signs of damage that cause changes in modal parameters are important in the TPS panel. When a sign of the damage was detected, the TPS panel would be inspected immediately by removing the TPS panel from the vehicle's structure. Therefore, we only focus on the damage presenting in the TPS panel rather than localizing exactly the position.

From the above-mentioned methods with 3DPT and MAC, we find that: (1) using 3DPT for detecting damage in a structure based on dynamic measurement, we can obtain more data compared with the traditional pointwise measurement method, and data storage and computation time are reduced compared with the full-field DIC method but precise mode shapes of the testing structure are still guaranteed. (2) Using MAC for damage detection, we can detect changes in global mode shapes of a testing structure. MAC is well suited for modal analysis of large-scale structures using a simultaneous measuring response method. Therefore, we employed the 3DPT method in conjunction with the MAC, which is a non-contact image-based method, to investigate dynamic characteristics, to obtain detailed mode shapes, and to identify damage states in the metallic TPS panel that are associated with the bolted joint loosening.

In this paper, we present a highly efficient 3DPT method in a conjunction with the MAC to investigate the dynamic characteristics of a complex structure, that is, a metallic TPS panel that consists of bolted joints, insulation layers, a load-carrying plate, and washers. We focus on detecting damage presenting in the TPS panel rather than localized damage. Three damage levels represented three cases of the number of damaged corners. For the experiment, we provided a discussion of good selections of the aperture size and suspension system by quantifying rigid body motions and rigid body natural frequencies. Then, an operational modal analysis (OMA), or ODS measurement during modal tests, was performed. The modal parameters, namely, the natural frequency, mode shape, and damping ratio, were identified with the aid of the OMA. The results of the 3DPT method have also been validated using the benchmark modal data from the accelerometers. The damage states were detected by comparing the natural frequencies and damping ratios with those of a healthy TPS panel. By taking advantage of the simultaneous multi-point measurement data, the ODSs were constructed and the modal matrix of the TPS panel was obtained from each measurement. The modal matrices of the first four modes were then processed to obtain MAC values which were used to identify the damage states. The signal-to-noise ratio of the captured images was too small in a certain area because of the limited resolution of cameras, intrinsic noise, low impact energy due to localized excitation, and infinitesimal deformation at the high frequency of the TPS panel. To remedy this drawback, a phase-based motion magnification was used. The results obtained from the 3DPT method demonstrated the advantage of using a non-wired setup and simultaneous measuring responses over the use of a single measuring response by an accelerometer. By using simultaneous multi-point measurement data for the MAC calculation, the 3DPT method reduced processing times significantly in the damage detection of a complex structure.

  1. Dorsey, J.; Poteet, C.; Chen, R.; Wurster, K. Metallic thermal protection system technology development-Concepts, requirements and assessment overview. In Proceedings of 40th AIAA Aerospace Sciences Meeting & Exhibit; p. 502.
  2. Le, V.T.; Goo, N.S. Thermomechanical performance of bio-inspired corrugated-core sandwich structure for a thermal protection system panel. Applied Sciences 2019, 9, 5541, doi:10.3390/app9245541.

3.Blosser, M.L.; Poteet, C.C.; Chen, R.R.; Dorsey, J.T.; Schmidt, I.H.; Bird, R.K.; Wurster, K.E. Development of advanced metallic-thermal-protection system prototype hardware. Journal of Spacecraft and Rockets 2004, 41, 183-194, doi:10.2514/1.9179.

  1. Xu, Y.; Xu, N.; Zhang, W.; Zhu, J. A multi-layer integrated thermal protection system with C/SiC composite and Ti alloy lattice sandwich. Composite Structures 2019, 10.1016/j.compstruct.2019.111507, 111507, doi:10.1016/j.compstruct.2019.111507.
  2. Li, Y.; Zhang, L.; He, R.; Ma, Y.; Zhang, K.; Bai, X.; Xu, B.; Chen, Y. Integrated thermal protection system based on C/SiC composite corrugated core sandwich plane structure. Aerospace Science and Technology 2019, 91, 607-616, doi:10.1016/j.ast.2019.05.048.
  3. Le, V.T.; Ha, N.S.; Goo, N.S. Thermal protective properties of the allomyrina dichotoma beetle forewing for thermal protection systems. Heat Transfer Engineering 2019, 40, 1539-1549, doi:10.1080/01457632.2018.1474603.
  4. Le, V.T.; Goo, N.S.; Kim, J.Y. Experimental investigation on thermal contact resistance of alumina fibrous insulation material with Ti-6Al-4V alloy at high temperature and its effective thermal conductivity. Heat and Mass Transfer 2019, 55, 1705-1721, doi:10.1007/s00231-018-02551-4.
  5. Le, V.T.; Ha, N.S.; Goo, N.S.; Kim, J.Y. Insulation system using high-temperature fibrous insulation materials. Heat Transfer Engineering 2019, 40, 1523-1538, doi:10.1080/01457632.2018.1474602.
  6. Blosser, M.L. Fundamental modeling and thermal performance issues for metallic thermal protection system concept. Journal of Spacecraft and Rockets 2004, 41, 195-206, doi:10.2514/1.9182.
  7. Le, V.T.; Goo, N.S.; Kim, J.Y. Thermomechanical behavior of superalloy thermal protection system under aerodynamic heating. Journal of Spacecraft and Rockets 2019, 56, 1432-1448, doi:10.2514/1.a34400.
  8. Ng, W.; McNamara, J.; Friedmann, P.; Waas, A. Thermomechanical behavior of damaged TPS including hypersonic flow effects. In Proceedings of 14th AIAA/AHI space planes and hypersonic systems and technologies conference; p. 7951.
  9. Ng, W.H.; Friedmann, P.; Waas, A. Thermomechanical analysis of a thermal protection system with defects and heat shorts. In Proceedings of 47th AIAA/ASME/ASCE/AHS/ASC Structures, Structural Dynamics, and Materials Conference 14th AIAA/ASME/AHS Adaptive Structures Conference 7th; p. 2212.
  10. Ng, W.; Friedmann, P.; Waas, A.; McNamara, J. Thermomechanical behavior of a thermal protection system with different levels of damage-experiments and simulation. In Proceedings of 48th AIAA/ASME/ASCE/AHS/ASC Structures, Structural Dynamics, and Materials Conference; p. 2272.
  11. Derriso, M.; Braisted, W.; Rosenstengel, J.; DeSimio, M. The structural health monitoring of a mechanically attached thermal protection system. JOM 2004, 56, 36-39, doi:10.1007/s11837-004-0030-9.
  12. Chen, R.R.; Blosser, M.L. Metallic thermal-protection-system panel flutter study. Journal of spacecraft and rockets 2004, 41, 207-212, doi:10.2514/1.9190.
  13. Tobe, R.J.; Grandhi, R.V. Hypersonic vehicle thermal protection system model optimization and validation with vibration tests. Aerospace Science and Technology 2013, 28, 208-213, doi:10.1016/j.ast.2012.11.001.
  14. Jiang, X.; Mahadevan, S.; Guratzsch, R. Bayesian wavelet methodology for damage detection of thermal protection system panels. AIAA journal 2009, 47, 942-952.
  15. Myers, D.E.; Martin, C.J.; Blosser, M.L. Parametric weight comparison of advanced metallic, ceramic tile, and ceramic blanket thermal protection systems; NASA TM 2000-210289; NASA: USA, 2000; p 49.
  16. Boehrk, H.; Weihs, H.; Elsäßer, H. Hot structure flight data of a faceted atmospheric reentry thermal protection system. International Journal of Aerospace Engineering 2019, 2019, 9754739, doi:10.1155/2019/9754739.
  17. Pau, A.; Greco, A.; Vestroni, F. Numerical and experimental detection of concentrated damage in a parabolic arch by measured frequency variations. Journal of Vibration and Control 2011, 17, 605-614.
  18. Yamaguchi, H.; Matsumoto, Y.; Kawarai, K.; Dammika, A.J.; Shahzad, S.; Takanami, R. Damage detection based on modal damping change in bridges. 2012.
  19. Farrar, C.R.; James Iii, G.H. System identification from ambient vibration measurements on a bridge. Journal of Sound and Vibration 1997, 205, 1-18, doi:10.1006/jsvi.1997.0977.
  20. Allemang, R.J. The modal assurance criterion–twenty years of use and abuse. Sound and vibration 2003, 37, 14-23.
  21. Orlowitz, E.; Brandt, A. Comparison of experimental and operational modal analysis on a laboratory test plate. Measurement 2017, 102, 121-130.
  22. Nguyen, M.; Filippatos, A.; Langkamp, A.; Gude, M. Modal identification of output-only systems of composite discs using Zernike modes and MAC. Sensors 2019, 19, 660, doi:10.3390/s19030660.
  23. Allemang, R.J.; Brown, D.L. A correlation coefficient for modal vector analysis. In Proceedings of Proceedings of the 1st international modal analysis conference; pp. 110-116.
  24. Zhong, H.; Yang, M. Damage detection for plate-like structures using generalized curvature mode shape method. Journal of Civil Structural Health Monitoring 2016, 6, 141-152, doi:10.1007/s13349-015-0148-1.
  25. Pandey, A.; Biswas, M. Damage detection in structures using changes in flexibility. Journal of sound and vibration 1994, 169, 3-17.
  26. Pandey, A.K.; Biswas, M.; Samman, M.M. Damage detection from changes in curvature mode shapes. Journal of Sound and Vibration 1991, 145, 321-332, doi:10.1016/0022-460X(91)90595-B.
  27. Altunışık, A.C.; Okur, F.Y.; Karaca, S.; Kahya, V. Vibration-based damage detection in beam structures with multiple cracks: modal curvature vs. modal flexibility methods. Nondestructive Testing and Evaluation 2019, 34, 33-53, doi:10.1080/10589759.2018.1518445.
  28. Sazonov, E.; Klinkhachorn, P. Optimal spatial sampling interval for damage detection by curvature or strain energy mode shapes. Journal of Sound and Vibration 2005, 285, 783-801.
  29. Capecchi, D.; Ciambella, J.; Pau, A.; Vestroni, F. Damage identification in a parabolic arch by means of natural frequencies, modal shapes and curvatures. Meccanica 2016, 51, 2847-2859.
  30. Bae, W.; Kyong, Y.; Dayou, J.; Park, K.-h.; Wang, S. Scaling the operating deflection shapes obtained from scanning laser doppler vibrometer. Journal of Nondestructive Evaluation 2011, 30, 91-98.
  31. Pai, P.F.; Young, L.G. Damage detection of beams using operational deflection shapes. International Journal of Solids and Structures 2001, 38, 3161-3192, doi:10.1016/S0020-7683(00)00274-2.
  32. Waldron, K.; Ghoshal, A.; Schulz, M.J.; Sundaresan, M.J.; Ferguson, F.; Pai, P.F.; Chung, J.H. Damage detection using finite element and laser operational deflection shapes. Finite Elements in Analysis and Design 2002, 38, 193-226, doi:10.1016/S0168-874X(01)00061-0.
  33. Xu, Y. Photogrammetry-based structural damage detection by tracking a visible laser line. Structural Health Monitoring 2019, 0, 1475921719840354, doi:10.1177/1475921719840354.
  34. Molina-Viedma, Á.; López-Alba, E.; Felipe-Sesé, L.; Díaz, F.; Rodríguez-Ahlquist, J.; Iglesias-Vallejo, M. Modal parameters evaluation in a full-scale aircraft demonstrator under different environmental conditions using HS 3D-DIC. Materials 2018, 11, 230.
  35. Civera, M.; Zanotti Fragonara, L.; Surace, C. Using video processing for the full-field identification of backbone curves in case of large vibrations. Sensors 2019, 19, 2345, doi:10.3390/s19102345.
  36. Civera, M.; Zanotti Fragonara, L.; Surace, C. Video Processing Techniques for the Contactless Investigation of Large Oscillations. Journal of Physics: Conference Series 2019, 1249, 012004, doi:10.1088/1742-6596/1249/1/012004.
  37. Tashan, J.; Al-Mahaidi, R. Detection of cracks in concrete strengthened with CFRP systems using infra-red thermography. Composites Part B: Engineering 2014, 64, 116-125, doi:10.1016/j.compositesb.2014.04.011.
  38. Liu, Y.; Yang, S.; Liu, X. Detection and quantification of damage in metallic structures by laser-generated ultrasonics. Applied Sciences 2018, 8, 824.
  39. Hwang, S.; An, Y.-K.; Yang, J.; Sohn, H. Remote Inspection of Internal Delamination in Wind Turbine Blades using Continuous Line Laser Scanning Thermography. International Journal of Precision Engineering and Manufacturing-Green Technology 2020, 7, 699-712, doi:10.1007/s40684-020-00192-9.
  40. Hu, Y.; Guo, W.; Zhu, W.; Xu, Y. Local damage detection of membranes based on Bayesian operational modal analysis and three-dimensional digital image correlation. Mechanical Systems and Signal Processing 2019, 131, 633-648, doi:10.1016/j.ymssp.2019.04.051.
  41. Ha, N.; Vang, H.; Goo, N. Modal analysis using digital image correlation technique: an application to artificial wing mimicking beetle’s hind wing. Experimental Mechanics 2015, 55, 989-998.
  42. Zhang, F.; Zarate Garnica, G.I.; Yang, Y.; Lantsoght, E.; Sliedrecht, H. Monitoring Shear Behavior of Prestressed Concrete Bridge Girders Using Acoustic Emission and Digital Image Correlation. Sensors 2020, 20, 5622, doi:10.3390/s20195622.
  43. Feito, N.; Calvo, J.V.; Belda, R.; Giner, E. An Experimental and Numerical Investigation to Characterize an Aerospace Composite Material with Open-Hole Using Non-Destructive Techniques. Sensors 2020, 20, 4148, doi:10.3390/s20154148.
  44. Zhao, T.; Le, V.T.; Goo, N.S. Global-local deformation measurement of stress concentration structures using a multi-digital image correlation system. Journal of Mechanical Science and Technology 2020, 34, 1655–1665, doi:10.1007/s12206-020-0328-8.
  45. Le, V.T.; Ha, N.S.; Jin, T.; Goo, N.S.; Kim, J.Y. Thermal interaction of a circular plate-ring structure using digital image correlation technique and infrared heating system. Journal of Mechanical Science and Technology 2016, 30, 4363-4372, doi:10.1007/s12206-016-0750-0.
  46. Ha, N.S.; Le, V.T.; Goo, N.S.; Kim, J.Y. Thermal strain measurement of austin stainless steel (ss304) during a heating-cooling process. International Journal of Aeronautical and Space Sciences 2017, 18, 206-214, doi:10.5139/ijass.2017.18.2.206.
  47. Jin, T.; Ha, N.S.; Le, V.T.; Goo, N.S.; Jeon, H.C. Thermal buckling measurement of a laminated composite plate under a uniform temperature distribution using the digital image correlation method. Composite Structures 2015, 123, 420-429, doi:10.1016/j.compstruct.2014.12.025.
  48. Pham, V.; Wang, H.; Xu, J.; Wang, J.; Singh, C.; Park, S. A Study of Substrate Models and Its Effect On Package Warpage Prediction. In Proceedings of 2019 IEEE 69th Electronic Components and Technology Conference (ECTC), 28-31 May 2019; pp. 1130-1139.
  49. Pham, V.-L.; Niu, Y.; Wang, J.; Wang, H.; Singh, C.; Park, S.; Zhong, C.; Koh, S.W.; Wang, J.; Shao, S. Experimentally Minimizing the Gap Distance Between Extra Tall Packages and PCB Using the Digital Image Correlation (DIC) Method. In Proceedings of 2018 IEEE 68th Electronic Components and Technology Conference (ECTC); pp. 1593-1599.
  50. Pham, V.-L.; Xu, J.; Pan, K.; Wang, J.; Park, S.; Singh, C.; Wang, H. Investigation of underfilling BGAs packages – Thermal fatigue. In Proceedings of 2020 IEEE 70th Electronic Components and Technology Conference (ECTC), Orlando, FL, USA; pp. 2252-2258.
  51. Ha, N.S.; Le, V.T.; Goo, N.S. Investigation of punch resistance of the Allomyrira dichtoloma beetle forewing. Journal of Bionic Engineering 2018, 15, 57-68, doi:10.1007/s42235-017-0004-6.
  52. Ha, N.S.; Le, V.T.; Goo, N.S. Investigation of fracture properties of a piezoelectric stack actuator using the digital image correlation technique. International Journal of Fatigue 2017, 101, 106-111, doi:10.1016/j.ijfatigue.2017.02.020.
  53. Srivastava, V.; Baqersad, J. An optical-based technique to obtain operating deflection shapes of structures with complex geometries. Mechanical Systems and Signal Processing 2019, 128, 69-81, doi:10.1016/j.ymssp.2019.03.021.
  54. Beberniss, T.J.; Ehrhardt, D.A. High-speed 3D digital image correlation vibration measurement: Recent advancements and noted limitations. Mechanical Systems and Signal Processing 2017, 86, 35-48, doi:10.1016/j.ymssp.2016.04.014.
  55. Wu, R.; Zhang, D.; Yu, Q.; Jiang, Y.; Arola, D. Health monitoring of wind turbine blades in operation using three-dimensional digital image correlation. Mechanical Systems and Signal Processing 2019, 130, 470-483, doi:10.1016/j.ymssp.2019.05.031.
  56. Helfrick, M.N.; Niezrecki, C.; Avitabile, P. Curvature methods of damage detection using digital image correlation. In Proceedings of Health Monitoring of Structural and Biological Systems 2009; p. 72950D.
  57. Poozesh, P.; Baqersad, J.; Niezrecki, C.; Avitabile, P.; Harvey, E.; Yarala, R. Large-area photogrammetry based testing of wind turbine blades. Mechanical Systems and Signal Processing 2017, 86, 98-115, doi:10.1016/j.ymssp.2016.07.021.

61. Warren, C.; Niezrecki, C.; Avitabile, P.; Pingle, P. Comparison of FRF measurements and mode shapes determined using optically image based, laser, and accelerometer measurements. Mechanical Systems and Signal Processing 2011, 25, 2191-2202, doi:10.1016/j.ymssp.2011.01.018.

Round 2

Reviewer 2 Report

The manuscript has been shortened as suggested, while also correcting where indicated and adding the missing details (e.g. about the Finite Element Analysis, the ME’scopeTM software, the ARAMIS software, and the Savitzky-Golay smoothing filter). The Authors also improved substantially their figures. In this sense, this Reviewer’s only remaining concern regards the grey-greenish colour of the lines in Figures 14, 17 (right side), and 18, which may still be a bit difficult to see. However, this is a minor point which can be handled by the Authors if they want to; the content of the paper does not require any further major revision.

The Authors corrected most of the typos throughout the manuscript; this Reviewer still recommends proper proofreading, possibly with an English native speaker, to be safe.

Author Response

Comments:

The manuscript has been shortened as suggested, while also correcting where indicated and adding the missing details (e.g. about the Finite Element Analysis, the ME’scopeTM software, the ARAMIS software, and the Savitzky-Golay smoothing filter). The Authors also improved substantially their figures. In this sense, this Reviewer’s only remaining concern regards the grey-greenish colour of the lines in Figures 14, 17 (right side), and 18, which may still be a bit difficult to see. However, this is a minor point which can be handled by the Authors if they want to; the content of the paper does not require any further major revision.

 The Authors corrected most of the typos throughout the manuscript; this Reviewer still recommends proper proofreading, possibly with an English native speaker, to be safe

Answer:

Thank you very much. We have changed the colors of the lines in the Figures mentioned by the reviewer. Please see the revised file.

We have tried to correct the typos throughout the manuscript. Most of the manuscript has been proofread by a native English corrector, who lives in San Francisco, USA. We would like to attach the proofreading confirmation in this note.

Again, we would like to thank the reviewer for your dedicated comments.

Reviewer 4 Report

Authors improved the paper according my review. I propose accept the paper in present  form.

Author Response

Thank you very much for your evaluation.